# Distances for Markov chains from sample streams

**Sergio Calo  Anders Jonsson  Gergely Neu  Ludovic Schwartz  Javier Segovia-Aguas**
Universitat Pompeu Fabra, Barcelona, Spain
`{sergio.calo,anders.jonsson,gergely.neu,ludovic.schwartz,javier.segovia}@upf.edu`

## Abstract

Bisimulation metrics are powerful tools for measuring similarities between stochastic processes, and specifically Markov chains. Recent advances have uncovered that bisimulation metrics are, in fact, optimal-transport distances, which has enabled the development of fast algorithms for computing such metrics with provable accuracy and runtime guarantees. However, these recent methods, as well as all previously known methods, assume full knowledge of the transition dynamics. This is often an impractical assumption in most real-world scenarios, where typically only sample trajectories are available. In this work, we propose a stochastic optimization method that addresses this limitation and estimates bisimulation metrics based on sample access, without requiring explicit transition models. Our approach is derived from a new linear programming (LP) formulation of bisimulation metrics, which we solve using a stochastic primal-dual optimization method. We provide theoretical guarantees on the sample complexity of the algorithm and validate its effectiveness through a series of empirical evaluations.

## 1   Introduction

Computing similarity metrics between stochastic processes is an important mathematical problem with numerous promising use cases in diverse areas such as mathematical finance, computational neuroscience, biology, and computer science. Within machine learning, potential applications include representation learning for dynamical systems and reinforcement learning [Zhang et al., 2021, Chen and Pan, 2022], fitting and comparing sequence models [Xu et al., 2020, Tao et al., 2024] or prediction tasks on graph-structured data [Titouan et al., 2019, Brugère et al., 2024]. While there exist several rigorous frameworks for defining such similarity metrics and studying their properties, computing them typically requires full knowledge of the probability law of the processes to compare, which is not available in just about any case of practical interest. In this paper, we address this problem by developing methods for estimating similarity metrics for a family of stochastic processes, based only on sample streams and without requiring any prior information about the underlying process laws.

We focus on a family of similarity metrics known as *bisimulation metrics*, originating from theoretical computer science for purposes of formal verification of computer programs [Park, 1981, Milner, 1989, Desharnais et al., 1999, van Breugel and Worrell, 2001]. This notion of process similarity has gained popularity within reinforcement learning (RL), where its potential for learning state representations has been recognized by the early works of Givan et al. [2003] and Ferns et al. [2004] and the possibility of using it as a basis of practical methods for representation learning has been explored in a long line of subsequent works [Castro, 2020, Zhang et al., 2021, Chen and Pan, 2022, Kemertas and Jepson, 2022]. Another popular framework for studying similarities between structured probability distributions is that of *optimal transport* (OT, cf. Villani, 2009), which has received serious attention within machine learning in the last decade, largely owing to the work of Cuturi [2013]. Very recently, Calo et al. [2024] pointed out that bisimulation metrics also fall within the family of OT distances, which not only allowed them to connect two distinct areas of mathematics but also import tools from the literature of computational optimal transport [Peyré and Cuturi, 2019]

39th Conference on Neural Information Processing Systems (NeurIPS 2025).

and develop more efficient methods for computing bisimulation metrics. We refer to Appendix A of Calo et al. [2024] for more historical details on the two extensive lines of literature on bisimulation metrics and optimal transport for stochastic processes.

In this paper, we extend this line of work and show that recasting bisimulation metrics as OT distances allows not only computational advances, but the development of a rigorous theory for *statistical estimation* of similarity metrics between stochastic processes. In particular, we build on the foundations laid down by Calo et al. [2024] and derive a new stochastic optimization algorithm for estimating bisimulation metrics based on sample observations only, and provide its complete computational and sample-complexity analysis for finite Markov chains. A core technical contribution is a new linear-program formulation of bisimulation metrics, which we solve via a stochastic saddle-point optimization method. For two Markov chains with state spaces $\mathcal{X}$ and $\mathcal{Y}$, the algorithm is guaranteed to return an $\varepsilon$-accurate estimate of the true similarity metric after $\widetilde{\mathcal{O}}(|\mathcal{X}| |\mathcal{Y}| (|\mathcal{X}| + |\mathcal{Y}|)/\varepsilon^2)$ iterations, with each iteration making use of a single sample transition from each of the two chains, and costing $\Theta(|\mathcal{X}|^2 |\mathcal{Y}|^2)$ computation. This is the first result of its kind: no previous methods have successfully addressed this problem either in practice or in theory.

As mentioned above, the problem we study in this paper has been extensively studied in (at least) two major research communities. Within the optimal-transport community, the problem of computing distances between stochastic processes has been studied under the names "adapted", "causal" or "bicausal" optimal transport [Pflug and Pichler, 2012, Lassalle, 2018, Backhoff-Veraguas et al., 2017, Eckstein and Pammer, 2024]. Considering the special case of Markov chains (as we do in the present paper), Moulos [2021], O'Connor et al. [2022] and Brugère et al. [2024] proposed approximate dynamic programming algorithms based on the observation that computing OT distances between Markov chains can be reduced to a problem of optimal control in Markov decision processes. Calo et al. [2024] developed a novel linear programming framework for computing OT distances between Markov chains, and showed that such distances are equivalent to bisimulation metrics. However, previous approaches in this line of work all assume known transition dynamics.

Within the theoretical computer science community, the study of bisimulation metrics progressed quite differently: after an initial flurry of foundational works of Desharnais et al. [1999], van Breugel and Worrell [2001] and Desharnais et al. [2002], surprisingly few studies have addressed computational matters (one rare example being the work of Chen et al. [2012]). Several recent works in reinforcement learning aim to learn approximate bisimulation metrics from sample transitions using deep learning [Castro, 2020, Zhang et al., 2021, Chen and Pan, 2022, Kemertas and Jepson, 2022]. However, these approximate bisimulation metrics are not well-founded in theory and as a consequence, do not enjoy the theoretical guarantees of the original metrics. In contrast, the stochastic-optimization method we develop in this paper is firmly rooted in a theoretical understanding of the problem and comes with provable computational and sample-complexity guarantees.

The use of stochastic solvers to compute OT distances has been explored in several past works, mostly in the context of static optimal transport between probability distributions. A good part of these methods are based on the observation that the static OT problem is formulated as a linear program, and the associated unconstrained dual optimization problem directly lends itself to numerical optimization. This view has been exploited by works like Genevay et al. [2016], Arjovsky et al. [2017], and Seguy et al. [2018], with some rigorous performance guarantees provided by Ballu et al. [2020]. Another line of work makes use of Monte Carlo estimates of OT distances [Genevay et al., 2018, Fatras et al., 2019, 2021, Mensch and Peyré, 2020]. To our knowledge, the idea of computing OT distances via stochastic primal-dual methods as we do in the present work has not been explored in this literature, and thus our contribution may be of independent interest within this context as well.

The rest of the paper is organized as follows. After formally defining our problem in Section 2, we describe the foundations of our new algorithm and describe it in detail in Section 3. We state our main theoretical results in Section 4, where we also outline the main ideas of the analysis. We complement these with some empirical studies of the newly proposed method in Section 5, and conclude with a discussion of the results and open problems in Section 6.

**Notation.** For a finite set $\mathcal{S}$, we use $\Delta_{\mathcal{S}}$ to denote the set of all probability distributions over $S$. For two sets $\mathcal{X}$ and $\mathcal{Y}$, we will often write $\mathcal{X}\mathcal{Y} = \mathcal{X} \times \mathcal{Y}$ to abbreviate their direct product. We will denote infinite sequences by $\bar{x} = (x_0, x_1, \ldots)$ and for any $n$ the corresponding subsequences as $\bar{x}_n = (x_0, \ldots, x_n)$.

## 2 Preliminaries

We study the problem of measuring distances between pairs of finite Markov chains. Specifically, we consider two stationary Markov processes $M_{\mathcal{X}} = (\mathcal{X}, P_{\mathcal{X}}, \nu_{0,\mathcal{X}})$ and $M_{\mathcal{Y}} = (\mathcal{Y}, P_{\mathcal{Y}}, \nu_{0,\mathcal{Y}})$, where

- $\mathcal{X}$ and $\mathcal{Y}$ are the *state spaces* with finite cardinality,
- $P_{\mathcal{X}} : \mathcal{X} \to \Delta_{\mathcal{X}}$ and $P_{\mathcal{Y}} : \mathcal{Y} \to \Delta_{\mathcal{Y}}$ are the *transition kernels* that specify the evolution of states as $P_{\mathcal{X}}(x'|x) = \mathbb{P}[X_{t+1} = x'|X_t = x]$ and $P_{\mathcal{Y}}(y'|y) = \mathbb{P}[Y_{t+1} = y'|Y_t = y]$ (for all time indices $t$ and state pairs $x, x'$ and $y, y'$),
- $\nu_{0,\mathcal{X}} \in \Delta(\mathcal{X})$ and $\nu_{0,\mathcal{Y}} \in \Delta(\mathcal{Y})$ are the *initial-state distributions* which specify the states at time $t = 0$ as $X_0 \sim \nu_{0,\mathcal{X}}$ and $Y_0 \sim \nu_{0,y}$.

Without loss of generality, we will assume that $\nu_{0,\mathcal{X}}$ and $\nu_{0,\mathcal{Y}}$ are both Dirac measures respectively supported on some fixed $x_0$ and $y_0$, and use $\nu_0 = \nu_{0,\mathcal{X}} \otimes \nu_{0,\mathcal{Y}}$ to denote the joint distribution of the pair of initial states $(X_0, Y_0)$ (which is of course a Dirac measure on $x_0, y_0$). For each $n \geq 0$, the objects above define a sequence of joint distributions $\mathbb{P}[(X_0, X_1, \ldots, X_n) = (x_0, x_1, \ldots, x_n)]$ and $\mathbb{P}[(Y_0, Y_1, \ldots, Y_n) = (y_0, y_1, \ldots, y_n)]$. These distributions in turn define the laws of the infinite sequences $\overline{X} = (X_0, X_1, \ldots)$ and $\overline{Y} = (Y_0, Y_1, \ldots)$ via Kolmogorov's extension theorem. With a slight abuse of notation we use $M_{\mathcal{X}}$ and $M_{\mathcal{Y}}$ to denote the corresponding measures satisfying $M_{\mathcal{X}}(\bar{x}_n) = \mathbb{P}[\overline{X}_n = \bar{x}_n]$ and $M_{\mathcal{Y}}(\bar{y}_n) = \mathbb{P}[\overline{Y}_n = \bar{y}_n]$ for any $\bar{x} \in \mathcal{X}^\infty, \bar{y} \in \mathcal{Y}^\infty$ and $n \geq 0$.

Our goal is to compute optimal transport distances between infinite-horizon Markov chains. To this end, we will suppose access to a *ground cost* (or *ground metric*) $c : \mathcal{X}\mathcal{Y} \to \mathbb{R}^+$ that quantifies the (dis-)similarity between each state $x \in \mathcal{X}$ and $y \in \mathcal{Y}$ as $c(x, y)$. For any two sequences $\bar{x} = (x_0, x_1, \ldots) \in \mathcal{X}^{\mathbb{N}}$ and $\bar{y} = (y_0, y_1, \ldots) \in \mathcal{Y}^{\mathbb{N}}$, we define the discounted total cost

$$c_\gamma(\bar{x}, \bar{y}) = \sum_{t=0}^\infty \gamma^t c(x_t, y_t),$$

where $\gamma \in (0, 1)$ is the *discount factor* (which emphasizes earlier differences between the two sequences, and serves to make sure that the distance is well-defined). As is usual in the optimal-transport literature, we will define the distance between the two stochastic processes $M_{\mathcal{X}}$ and $M_{\mathcal{Y}}$ by minimizing the expected cost over a suitable class of *couplings* of the two joint distributions.

Formally, a coupling of $M_{\mathcal{X}}$ and $M_{\mathcal{Y}}$ is defined as a stochastic process on the joint space $\mathcal{X} \times \mathcal{Y}$ whose law is defined for all $n$ as $M_{\mathcal{X},\mathcal{Y}}(\bar{x}_n, \bar{y}_n) = \mathbb{P}[\overline{X}_n = x_n, \overline{Y}_n = y_n]$ and satisfies $\sum_{\bar{y}_n \in \mathcal{Y}^n} M_{\mathcal{X},\mathcal{Y}}(\bar{x}_n \bar{y}_n) = M_{\mathcal{X}}(\bar{x}_n)$ and $\sum_{\bar{x}_n \in \mathcal{X}^n} M_{\mathcal{X},\mathcal{Y}}(\bar{x}_n, \bar{y}_n) = M_{\mathcal{Y}}(\bar{y}_n)$. We denote the set of all couplings by $\Pi$, and call a coupling $M_{\mathcal{X},\mathcal{Y}} \in \Pi$ *bicausal* if and only if it satisfies

$$\sum_{y \in \mathcal{Y}} M_{\mathcal{X},\mathcal{Y}}(xy|\bar{x}_{n-1}\bar{y}_{n-1}) = M_{\mathcal{X}}(x|\bar{x}_{n-1}) \quad \text{and} \quad \sum_{x \in \mathcal{X}} M_{\mathcal{X},\mathcal{Y}}(xy|\bar{x}_{n-1}\bar{y}_{n-1}) = M_{\mathcal{Y}}(y|\bar{y}_{n-1}),$$

respectively for all $x$ and $y$, and for all $n$. The set of all bicausal couplings will be denoted by $\Pi_{bc}$. Intuitively, this is the class of couplings that respect the temporal structure of the Markov chains and only allow the distribution of each state $X_{t+1}$ (resp. $Y_{t+1}$) to be influenced by the past state pairs $(\overline{X}_t, \overline{Y}_t)$. The optimal transport distance between the two Markov chains $M_{\mathcal{X}}$ and $M_{\mathcal{Y}}$ is then defined as

$$d_\gamma(M_{\mathcal{X}}, M_{\mathcal{Y}}) = \inf_{\pi \in \Pi_{bc}} \int c_\gamma(\overline{X}, \overline{Y}) \, d\pi(\overline{X}, \overline{Y}), \tag{1}$$

with the dependence on the cost function $c$ suppressed for simplicity of notation. Following the observation made by Calo et al. [2024], we will frequently refer to this distance as the *bisimulation metric* between $M_{\mathcal{X}}$ and $M_{\mathcal{Y}}$.

## 3 Bisimulation metrics from sample streams

As observed by Calo et al. [2024], the bisimulation metric in (1) can be rewritten in terms of *occupancy couplings*. The occupancy coupling associated with the bicausal coupling $\pi \in \Pi_{bc}$ is a distribution $\mu^\pi \in \Delta_{\mathcal{X}\mathcal{Y}\mathcal{X}\mathcal{Y}}$ with entries

$$\mu^\pi(x, y, x', y') = (1 - \gamma) \sum_{t=0}^\infty \gamma^t \mathbb{P}_\pi[X_t = x, Y_t = y, X_{t+1} = x', Y_{t+1} = y'],$$

where $\mathbb{P}_\pi[\cdot]$ denotes the probability law induced by the coupling $\pi$. Introducing the notation $\langle\mu, c\rangle = \sum_{x,y,x',y'} \mu(x,y,x',y')c(x,y)$, this means that the original optimization problem defining the distance can be obviously rewritten as a linear function of $\mu^\pi$ as

$$d_\gamma(M_\mathcal{X}, M_\mathcal{Y}) = \inf_{\pi\in\Pi_{bc}} \langle\mu^\pi, c\rangle. \qquad (2)$$

Calo et al. [2024] identified a set of linear constraints on $\mu^\pi$ that are satisfied if and only if $\pi \in \Pi_{bc}$, which has effectively reduced the problem of computing the distance to a linear program (LP). This formulation is closely related to the standard LP formulation of optimal control in Markov decision processes, where the primal variables are commonly called *occupancy measures* (see, e.g., Chapter 6.9 in Puterman, 1994). As shown by Calo et al. [2024], one of the linear constraints satisfied by any valid occupancy $\mu$ is the following *flow condition*:

$$\sum_{x',y'} \mu(x,y,x',y') = \gamma \sum_{\widehat{x},\widehat{y}} \mu(\widehat{x},\widehat{y},x,y) + (1-\gamma)\nu_0(x,y) \qquad (\forall x, y \in \mathcal{X}\mathcal{Y}). \qquad (3)$$

Unfortunately, their other constraints explicitly feature the transition kernels $P_\mathcal{X}$ and $P_\mathcal{Y}$, which ultimately makes their LP unsuitable as a basis for stochastic optimization. Indeed, optimizing their LP via primal, dual, or primal-dual methods would require having at least a generative model of $P_\mathcal{X}$ and $P_\mathcal{Y}$ that allows sampling from $P_\mathcal{X}(\cdot|x)$ and $P_\mathcal{Y}(\cdot|y)$ at arbitrary states $x$ and $y$. In practice however, such models are rarely available and one has to make do with samples drawn directly from a stream of states generated by the two chains. We address this limitation by reformulating their linear constraints in a form that eliminates the transition kernels $P_\mathcal{X}$ and $P_\mathcal{Y}$ from the constraints, and replaces them with a joint state-next-state distribution from each chain that can be sampled from efficiently. In what follows, we first introduce our new LP formulation, and then provide a primal-dual stochastic optimization algorithm to approximately solve the resulting optimization problem along with its performance guarantees.

### 3.1 A new LP formulation of bisimulation metrics

Our reformulation is based on the following observations. First, notice that any valid occupancy coupling has to arise as a coupling of the marginal occupancy measures of the two chains $M_\mathcal{X}$ and $M_\mathcal{Y}$, defined respectively for each $x, x'$ and $y, y'$ as

$$\nu_\mathcal{X}(x,x') = (1-\gamma)\sum_{t=0}^\infty \gamma^t\mathbb{P}[X_t = x, X_{t+1} = x'],$$
$$\nu_\mathcal{Y}(y,y') = (1-\gamma)\sum_{t=0}^\infty \gamma^t\mathbb{P}[Y_t = y, Y_{t+1} = y'].$$

Indeed, valid occupancy couplings respectively satisfy the *coupling condition*

$$\sum_{y,y'} \mu^\pi(x,x',y,y') = \nu_\mathcal{X}(x,x') \qquad \text{and} \qquad \sum_{x,x'} \mu^\pi(x,x',y,y') = \nu_\mathcal{Y}(y,y') \qquad (4)$$

for all $x, x'$ and $y, y'$. Second, the conditional occupancies induced by a bicausal coupling $\pi$ satisfy

$$\mu^\pi(x',y|x) = P_\mathcal{X}(x'|x)\mu^\pi(y|x) \qquad \text{and} \qquad \mu^\pi(x,y'|y) = P_\mathcal{Y}(y'|y)\mu^\pi(x|y),$$

due to the requirement of *causality* that the conditional law of $Y_t$ given $X_t$ (resp. $X_t$ given $Y_t$) should be independent of the future state $X_{t+1}$ (resp. $Y_{t+1}$). By multiplying both sides of these equations by $\sum_{x'} \nu_\mathcal{X}(x,x')$ and $\sum_{y'} \nu_\mathcal{Y}(y,y')$, we obtain

$$\sum_{y'} \mu^\pi(x,x',y,y') = \nu_\mathcal{X}(x,x')\mu^\pi(y|x) \quad \text{and} \quad \sum_{x'} \mu^\pi(x,x',y,y') = \nu_\mathcal{Y}(y,y')\mu^\pi(x|y). \qquad (5)$$

Summing both sides for all $y$ and $x$ respectively recovers the coupling conditions of Equation (4). In this sense, both the causality and coupling conditions can be recovered by the single set of equations (5). The following key result shows that, together with the flow constraints of Equation (3), this system of equations provides a complete characterization of occupancy couplings.

**Proposition 1.** *The distribution $\mu$ is the induced occupancy coupling of a bicausal coupling $\pi \in \Pi_{bc}$ if and only if there exist $\lambda_\mathcal{X} \in \mathbb{R}_+^{\mathcal{Y}\mathcal{X}}$ and $\lambda_\mathcal{Y} \in \mathbb{R}_+^{\mathcal{X}\mathcal{Y}}$ such that the following equations hold:*

$$\sum_{x',y'} \mu(x,y,x',y') = \gamma \sum_{\widehat{x},\widehat{y}} \mu(\widehat{x},\widehat{y},x,y) + (1-\gamma)\nu_0(x,y) \qquad (\forall x, y \in \mathcal{X}\mathcal{Y}) \qquad (6)$$

$$\sum_{y'} \mu(x,y,x',y') = \nu_\mathcal{X}(x,x')\lambda_\mathcal{X}(y|x) \qquad (\forall x, x', y \in \mathcal{X}\mathcal{X}\mathcal{Y}) \qquad (7)$$

$$\sum_{x'} \mu(x,y,x',y') = \nu_\mathcal{Y}(y,y')\lambda_\mathcal{Y}(x|y) \qquad (\forall x, y, y' \in \mathcal{X}\mathcal{Y}\mathcal{Y}). \qquad (8)$$

*Furthermore, if the equations are satisfied for some $\mu$, $\lambda_{\mathcal{X}}$ and $\lambda_{\mathcal{Y}}$, we also have $\sum_y \lambda_{\mathcal{X}}(y|x) = 1$ and $\sum_x \lambda_{\mathcal{Y}}(x|y) = 1$ for all $x$ and $y$.*

Thus, the set of equations (6)–(8) uniquely identifies the complete set of occupancy couplings. In particular, whenever the constraints are satisfied for some $\mu$, there exists a bicausal coupling $\pi$ inducing $\mu$ as its occupancy coupling, and conversely all occupancy couplings satisfy the above equations. Further important side results can be read out from the proof, provided in Appendix A.

## 3.2 A stochastic primal-dual method

An immediate consequence of Proposition 1 is that the OT distance between $M_{\mathcal{X}}$ and $M_{\mathcal{Y}}$ can be written as the solution of the minimization problem of Equation (2) with respect to $\mu^\pi$ as the optimization variable, subject to the constraints (6)–(8). Equivalently, it can be written as a saddle point of the associated Lagrangian defined as

$$\mathcal{L}(\mu, \lambda; \alpha, V) = \sum_{xyx'y'} \mu(x, y, x', y') \left( c(x, y) + \alpha_{\mathcal{X}}(x, x', y) + \alpha_{\mathcal{Y}}(x, y, y') + \gamma V(x', y') - V(x, y) \right)$$

$$- \sum_{xx'y} \nu_{\mathcal{X}}(x, x') \lambda_{\mathcal{X}}(y|x) \alpha_{\mathcal{X}}(x, x', y) - \sum_{xyy'} \nu_{\mathcal{Y}}(y, y') \lambda_{\mathcal{Y}}(x|y) \alpha_{\mathcal{Y}}(x, y, y')$$

$$+ (1 - \gamma) \sum_{xy} \nu_0(x, y) V(x, y), \tag{9}$$

where $\alpha_{\mathcal{X}} \in \mathbb{R}^{\mathcal{X} \mathcal{X} \mathcal{Y}}$ and $\alpha_{\mathcal{Y}} \in \mathbb{R}^{\mathcal{X} \mathcal{Y} \mathcal{Y}}$ are the Lagrange multipliers associated with constraints (7) and (8), and $V \in \mathbb{R}^{\mathcal{X} \mathcal{Y}}$ are the multipliers for the flow constraint (6). Indeed, by the Lagrange multiplier theorem, the optimal value of the original LP can be written as $d_\gamma(M_{\mathcal{X}}, M_{\mathcal{Y}}) = \min_{\mu, \lambda} \max_{\alpha, V} \mathcal{L}(\mu, \lambda; \alpha, V)$. Importantly, the gradients of the Lagrangian with respect to dual variables $\alpha_{\mathcal{X}}$ and $\alpha_{\mathcal{Y}}$ can be written as expectations with respect to the occupancy measures $\nu_{\mathcal{X}}$ and $\nu_{\mathcal{Y}}$, which suggests that the objective may be amenable to stochastic optimization given sample access to these distributions.

Inspired by this observation, we propose a primal-dual stochastic optimization algorithm that aims to approximate the saddle point of the Lagrangian. In particular, we will suppose that we have sampling access to the occupancy measures $\nu_{\mathcal{X}}$ and $\nu_{\mathcal{Y}}$ and use these samples to construct stochastic gradient estimators for incrementally updating the primal and dual variables via variants of stochastic gradient descent-ascent. Concretely, the algorithm proceeds in a sequence of iterations $k = 1, 2, \ldots, K$, updating the primal variables $\mu_k$, $\lambda_{\mathcal{X},k}$ and $\lambda_{\mathcal{Y},k}$ via

---

**Algorithm 1 SOMCOT**

**Input:** $c, \eta, \beta, \gamma, K$
**Initialize:** $\mu_1 = \mathcal{U}(\mathcal{X}\mathcal{Y}\mathcal{X}\mathcal{Y})$, $\lambda_{\mathcal{X}}(\cdot|x) = \mathcal{U}(\mathcal{Y})$ for all $x$, $\lambda_{\mathcal{Y}}(\cdot|y) = \mathcal{U}(\mathcal{X})$ for all $y$, $\alpha = 0$, $V = 0$.
**For** $k = 1, 2, \ldots, K$:
• Sample $X_k, X_k' \sim \nu_{\mathcal{X}}$ and $Y_k, Y_k' \sim \nu_{\mathcal{Y}}$,
• compute gradient estimators via Eqs. (10)–(15),
• update primal parameters via Eqs. (16)–(18),
• update dual parameters via Eqs. (19)–(21).
**Output:** $\overline{\mu}_K = \frac{1}{K} \sum_{k=1}^{K} \mu_k$.

---

stochastic mirror descent (SMD) with entropic regularization and the dual variables $V_k$, $\alpha_{\mathcal{X},k}$ and $\alpha_{\mathcal{Y},k}$ via stochastic gradient ascent (SGA). We describe the gradient-estimation procedures and the update rules below, and provide a high-level pseudocode as Algorithm 1. For brevity, we will refer to the algorithm as **SOMCOT**, for Stochastic Optimization for Markov Chain Optimal Transport. Further details about the derivation of **SOMCOT** and a more detailed pseudocode can be found in Appendix B.

**The gradient estimators.** For constructing the gradient estimators needed for the updates, we first sample transitions $(X_k, X_k') \sim \nu_{\mathcal{X}}$ and $(Y_k, Y_k') \sim \nu_{\mathcal{Y}}$ from the marginal occupancy measures of $M_{\mathcal{X}}$ and $M_{\mathcal{Y}}$, and let $\mathcal{F}_k$ denote the record of all transition data drawn until the end of round $k$. The primal updates are defined in terms of the following gradient estimates:

$$g_{k,\mu}(x, y, x', y') = c(x, y) - \alpha_{\mathcal{X},k}(x, x', y) - \alpha_{\mathcal{Y},k}(x, y, y') + \gamma V_k(x', y') - V_k(x, y) \tag{10}$$

$$\widetilde{g}_{k,\lambda_{\mathcal{X}}}(y|x) = \mathbf{1}_{\left\{X_k, X_k' = x, x'\right\}} \alpha_{\mathcal{X},k}(x, x', y) \tag{11}$$

$$\widetilde{g}_{k,\lambda_{\mathcal{Y}}}(x|y) = \mathbf{1}_{\left\{Y_k, Y_k' = y, y'\right\}} \alpha_{\mathcal{Y},k}(x, y, y'). \tag{12}$$

Clearly, we have $g_{k,\mu} = \nabla_\mu \mathcal{L}(\mu_k, \lambda_k; \alpha_k, V_k)$. Furthermore, it is easy to check that $\mathbb{E}\left[\widetilde{g}_{k,\lambda_{\mathcal{X}}} | \mathcal{F}_{k-1}\right] = \nabla_{\lambda_{\mathcal{X}}} \mathcal{L}(\mu_k, \lambda_k; \alpha_k, V_k)$ and $\mathbb{E}\left[\widetilde{g}_{k,\lambda_{\mathcal{Y}}} | \mathcal{F}_{k-1}\right] = \nabla_{\lambda_{\mathcal{Y}}} \mathcal{L}(\mu_k, \lambda_k; \alpha_k, V_k)$. Simi-

larly, we can define the gradient estimates for the dual variables as

$$\widetilde{g}_{k,\alpha_{\mathcal{X}}}(x, x', y) = \sum_{y'} \mu_k(x, y, x', y') - \mathbf{1}_{\{X_k, X'_k = x, x'\}} \lambda_{\mathcal{X},k}(y|x) \tag{13}$$

$$\widetilde{g}_{k,\alpha_{\mathcal{Y}}}(x, y, y') = \sum_{x'} \mu_k(x, y, x', y') - \mathbf{1}_{\{Y_k, Y'_k = y, y'\}} \lambda_{\mathcal{Y},k}(x|y) \tag{14}$$

$$g_{k,V}(x, y) = \sum_{x'y'} \mu_k(x, y, x', y') - (1 - \gamma)\nu_0(x, y) - \gamma \sum_{\widehat{x},\widehat{y}} \mu_k(\widehat{x}, \widehat{y}, x, y), \tag{15}$$

which are again easily seen to satisfy $\mathbb{E}\left[\widetilde{g}_{k,\alpha_{\mathcal{X}}} \mid \mathcal{F}_{k-1}\right] = \nabla_{\alpha_{\mathcal{X}}}\mathcal{L}(\mu_k, \lambda_k; \alpha_k, V_k)$, $\mathbb{E}\left[\widetilde{g}_{k,\alpha_{\mathcal{Y}}} \mid \mathcal{F}_{k-1}\right] = \nabla_{\alpha_{\mathcal{Y}}}\mathcal{L}(\mu_k, \lambda_k; \alpha_k, V_k)$ and $g_{k,V} = \nabla_V \mathcal{L}(\mu_k, \lambda_k; \alpha_k, V_k)$.

**The update rules.** The primal variables are updated via stochastic mirror descent with appropriately chosen entropic regularization functions. For $\mu$, the updates are given as

$$\mu_{k+1}(x, y, x', y') = \frac{\mu_k(x, y, x', y') \exp(-\eta g_{k,\mu}(x, y, x', y'))}{\sum_{\widehat{x}\widehat{y}\widehat{x}'\widehat{y}'} \mu_k(\widehat{x}, \widehat{y}, \widehat{x}', \widehat{y}') \exp(-\eta g_{k,\mu}(\widehat{x}, \widehat{y}, \widehat{x}', \widehat{y}'))}, \tag{16}$$

with $\eta > 0$ being a stepsize parameter, and the $\lambda_{\mathcal{X}}$ variables are updated as

$$\lambda_{\mathcal{X},k+1}(y|x) = \frac{\lambda_{\mathcal{X},k}(y|x) \exp(-\eta_{\mathcal{X}} \widetilde{g}_{k,\lambda_{\mathcal{X}}}(y|x))}{\sum_{\widehat{y}} \lambda_{\mathcal{X},k}(\widehat{y}|x) \exp(-\eta_{\mathcal{X}} \widetilde{g}_{k,\lambda_{\mathcal{X}}}(\widehat{y}|x))}, \tag{17}$$

$$\lambda_{\mathcal{Y},k+1}(x|y) = \frac{\lambda_{\mathcal{Y},k}(x|y) \exp(-\eta_{\mathcal{Y}} \widetilde{g}_{k,\lambda_{\mathcal{Y}}}(x|y))}{\sum_{\widehat{x}} \lambda_{\mathcal{Y},k}(\widehat{x}|y) \exp(-\eta_{\mathcal{Y}} \widetilde{g}_{k,\lambda_{\mathcal{Y}}}(\widehat{x}|y))}, \tag{18}$$

with respective stepsize parameters $\eta_{\mathcal{X}}, \eta_{\mathcal{Y}} > 0$. Note that due to the design of the gradient estimators, each iteration only needs to update these variables locally at $\lambda_{\mathcal{X}}(\cdot|X_k)$ and $\lambda_{\mathcal{Y}}(\cdot|Y_k)$ at the sampled states $X_k$ and $Y_k$. For the dual variables, we define $\Pi_{\mathcal{D}}$ as the orthogonal projection operator onto a convex set $\mathcal{D}$, and implement the following projected stochastic gradient ascent updates:

$$\alpha_{\mathcal{X},k+1} = \Pi_{\mathcal{D}_\alpha}\left[\alpha_{\mathcal{X},k} - \beta_{\mathcal{X}}\widetilde{g}_{k,\alpha_{\mathcal{X}}}\right], \tag{19}$$

$$\alpha_{\mathcal{Y},k+1} = \Pi_{\mathcal{D}_\alpha}\left[\alpha_{\mathcal{Y},k} - \beta_{\mathcal{Y}}\widetilde{g}_{k,\alpha_{\mathcal{Y}}}\right], \tag{20}$$

$$V_{k+1} = \Pi_{\mathcal{D}_V}\left[V_k - \beta g_{k,V}\right], \tag{21}$$

where $\mathcal{D}_V = \mathcal{B}^\infty(0, \frac{2}{1-\gamma})$ and $\mathcal{D}_\alpha = \mathcal{B}^\infty(0, \frac{6}{1-\gamma})$ are the projection domains for each variable, and $\beta, \beta_{\mathcal{X}}, \beta_{\mathcal{Y}} > 0$ are the stepsize parameters.

**The output.** The algorithm terminates after $K$ rounds, and produces the average of the primal iterates $\overline{\mu}_K = \frac{1}{K}\sum_{k=1}^K \mu_k$ as output. From this, an estimate of the distance can be computed as $\widehat{d}_\gamma(M_{\mathcal{X}}, M_{\mathcal{Y}}) = \langle\overline{\mu}_K, c\rangle$. Averaging the output variables is motivated by the design of **SOMCOT** as a primal-dual method and its theoretical analysis, and it also helps stabilize the quality of the solution. Indeed, primal-dual methods are prone to instability and oscillations, which are smoothed out very effectively by averaging. We discuss the role of this step and other practical improvements to the algorithm in Section 5 below, and provide further comments on the potential usefulness of other side products computed by **SOMCOT** for downstream tasks.

## 4  Analysis

The following theorem is our main theoretical result about the performance of our algorithm.

**Theorem 1.** *Suppose that $\|c\|_\infty \leq 1$. Let $\overline{\mu}_K = \frac{1}{K}\sum_{k=1}^K \mu_k$ be the output of **SOMCOT**, let $\mu^*$ be the optimal occupancy coupling achieving the minimum in Equation* (2)*, and set the learning rates as*

$$\eta = \sqrt{\frac{\log(|\mathcal{X}|^2\,|\mathcal{Y}|^2)(1-\gamma)^2}{K}}, \quad \eta_{\mathcal{X}} = \sqrt{\frac{|\mathcal{X}|\log|\mathcal{Y}|\,(1-\gamma)^2}{K}}, \quad \eta_{\mathcal{Y}} = \sqrt{\frac{|\mathcal{Y}|\log|\mathcal{X}|\,(1-\gamma)^2}{K}},$$

$$\beta_{\mathcal{X}} = \sqrt{\frac{|\mathcal{X}|^2\,|\mathcal{Y}|}{(1-\gamma)^2 K}}, \quad \beta_{\mathcal{Y}} = \sqrt{\frac{|\mathcal{X}|\,|\mathcal{Y}|^2}{(1-\gamma)^2 K}}, \quad \beta = \sqrt{\frac{|\mathcal{X}|\,|\mathcal{Y}|}{(1-\gamma)^2 K}}.$$

*Then, the following bound is satisfied with probability at least $1 - \delta$:*

$$|\langle \overline{\mu}_K - \mu^*, c \rangle| = \mathcal{O}\left( \frac{1}{\sqrt{K}(1-\gamma)} \left( \sqrt{|\mathcal{X}|\,|\mathcal{Y}|\,(|\mathcal{X}| + |\mathcal{Y}|)} + \sqrt{(|\mathcal{X}| + |\mathcal{Y}|)\log{(1/\delta)}} \right) \right).$$

*Equivalently, for any $\varepsilon > 0$, the output satisfies $|\langle \overline{\mu}_K - \mu^*, c \rangle| \leq \varepsilon$ with probability at least $1 - \delta$ if the number of iterations is at least $K \geq K_0 = \mathcal{O}\left( \frac{|\mathcal{X}||\mathcal{Y}|(|\mathcal{X}|+|\mathcal{Y}|)+(|\mathcal{X}|+|\mathcal{Y}|)\log(1/\delta)}{(1-\gamma)^2\varepsilon^2} \right)$.*

A perhaps surprising feature of the sample-complexity guarantee is that it scales with the state spaces as $|\mathcal{X}|\,|\mathcal{Y}|\,(|\mathcal{X}| + |\mathcal{Y}|)$ instead of the full dimensionality of the decision variables, $|\mathcal{X}|^2\,|\mathcal{Y}|^2$. Note however that each iteration has a computational cost scaling with this full dimensionality. The scaling in terms of $\varepsilon$ is optimal up to logarithmic factors, as can be deduced from well-known lower bounds for the static OT problem (see, e.g., Klatt et al. 2020). Finally, we note that the big-O notation only hides numerical constants, and the bound features no problem-dependent factors whatsoever.

We provide the main idea of the proof below, and relegate the full analysis to Appendix C. The main technical idea is to relate the estimated transport cost $\langle \mu_K, c \rangle$ to the true optimal transport cost via the analysis of the *duality gap* associated with the sequence of iterates computed by the algorithm. The duality gap $\mathcal{G}_K(\mu^*, \lambda^*; \alpha^*, V^*)$ against a set of comparator points $(\mu^*, \lambda^*; \alpha^*, V^*)$ satisfies

$$\mathcal{G}_K(\mu^*, \lambda^*; \alpha^*, V^*) = \frac{1}{K}\sum_{k=1}^{K} \left( \mathcal{L}(\mu_k, \lambda_k; \alpha^*, V^*) - \mathcal{L}(\mu^*, \lambda^*; \alpha_k, V_k) \right). \tag{22}$$

As is standard for analysis of primal-dual methods, the duality gap can be decomposed into the sum of the *regrets* of the minimizing player controlling $\mu$ and $\lambda$, and the maximizing player controlling $\alpha$ and $V$, which can be controlled using the well-established of online learning [Cesa-Bianchi and Lugosi, 2006, Orabona, 2019]. For the analysis, we will pick the comparator points as follows. For the primal variables, we let $\mu^*$ be the occupancy coupling achieving the minimum in Equation (2) and let the $\lambda^*$ variables be the conditional distributions of $Y|X$ and $X|Y$ under the joint distribution $\mu^*$. For the dual variables, we choose

$$(\alpha^*, V^*) = \underset{\alpha \in \mathcal{D}_\alpha, V \in \mathcal{D}_V}{\arg\max} \; \frac{1}{K}\sum_{k=1}^{K} \mathcal{L}(\mu_k, \lambda_k; \alpha, V).$$

Under these choices, the error can be upper bounded as follows.

**Lemma 1.** $|\langle \overline{\mu}_K - \mu^*, c \rangle| \leq \mathcal{G}_K(\mu^*, \lambda^*, \alpha^*, V^*).$

The proof of this lemma makes up the bulk of the analysis, and is thus relegated to Appendix C.1. It then remains to upper-bound the regrets of the two sets of players, which is routine work that we execute in Appendix C.3.

## 5   Experiments

We performed a suite of numerical experiments to study the empirical behavior of our newly proposed algorithm, as well as to illustrate some potential applications that are enabled by our method. Due to space restrictions, we only show a small portion of the results here, and refer the reader to Appendix F for additional results and implementation details (most notably a detailed discussion on hyperparameter-tuning).

Several of the experiments are conducted with a family of processes we call *block Markov chains*, motivated by the framework of block Markov decision processes (or block MDPs, Du et al. 2019). This framework is commonly studied in the context of representation learning for reinforcement learning, where a standard postulate is that the dynamics of the environment are governed by a simple latent structure. Block Markov chains formalize this setting by assuming the existence of a latent Markov chain with a small discrete state space, with each latent state generating a unique set of observations. Formally, we emulate the block structure by fixing a low-dimensional chain $M_\mathcal{X}$ and another chain $M_\mathcal{Y}$ that is a copy of $M_\mathcal{X}$ up to an additional irrelevant noise variable. In our experiments, we let $M_\mathcal{X}$ be a uniform random walk on the state space $\mathcal{X} = \{1, 2, \ldots, n\}$ and $M_\mathcal{Y}$ is a Markov chain on the state space $\mathcal{X} \times \{1, 2, \ldots, B\}$, with the value in $\{1, 2, \ldots, B\}$ generated uniformly at random. In all experiments, we use a sparse cost function that only allows to clearly distinguish between states $x = 1$ and $x = n$, and treats all other states as identical.

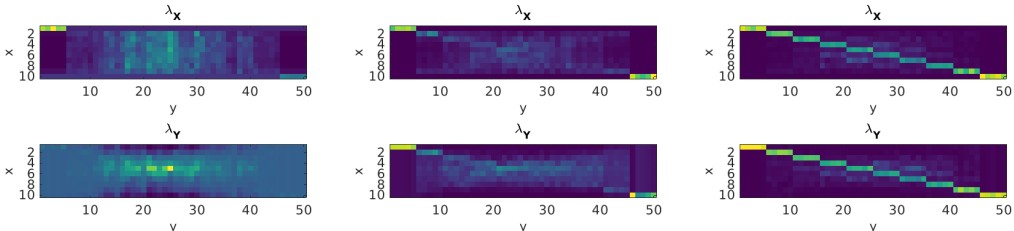

Figure 1: Encoder-decoder maps learned by the algorithm in a block Markov chain example ($n = 10$, $B = 5$) for sample sizes 1000, 10000 and 100000.

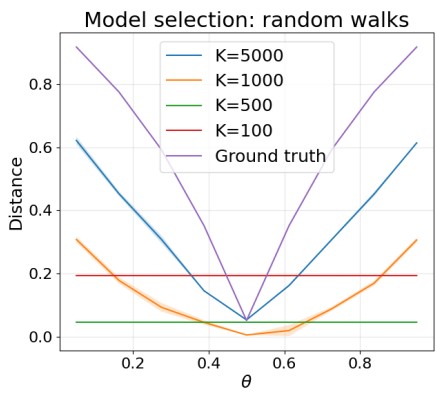

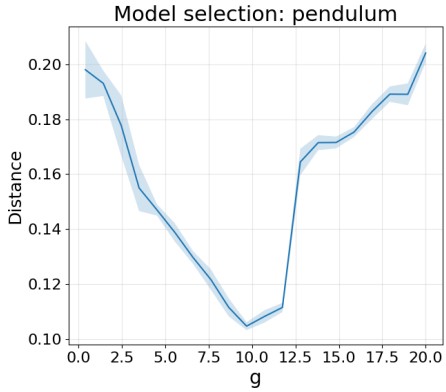

(a) Distances between the true dynamics ($\theta = 0.5$) and the different models in the model class, for different values of the parameter $\theta$ and the number of iterations $K$.

(b) Distances between the true dynamics ($g = 9.8$) and the different models in the model class, for different values of the gravity parameter $g$.

Figure 2: Model selection results for random walks and the pendulum environment

**Representation learning.** Within the family of block Markov processes, the task of representation learning is equivalent to finding the mapping between the latent states and the observations and vice versa. Our method is very well suited for this task, thanks to the following curious observation. Besides the estimated coupling $\overline{\mu}_K$ and the associated cost, the algorithm outputs other values that are potentially useful. Among these, the variables $\overline{\lambda}_{\mathcal{X}} = \frac{1}{K} \sum_{k=1}^{K} \lambda_{\mathcal{X},k}$ and $\overline{\lambda}_{\mathcal{Y}} = \frac{1}{K} \sum_{k=1}^{K} \lambda_{\mathcal{Y},k}$ are particularly interesting for purposes of representation, as these conditional distributions can be interpreted as an *encoder-decoder* pair, with $\lambda_{\mathcal{X}}(\cdot|x)$ and $\lambda_{\mathcal{Y}}(\cdot|y)$ giving the respective conditional distributions of $Y|X = x$ and $X|Y = y$ under the estimate of the optimal coupling. To illustrate the potential usefulness of these maps, we conducted a set of experiments on block Markov chains with parameters $n = 10$ and $B = 5$, and show the encoder-decoder pairs computed by **SOMCOT** in Figure 1. Notably, the algorithm does not make use of any prior structural knowledge of the environment: each individual state $y$ is treated as a separate state. Despite this and the very limited information revealed by the cost function, a block structure is clearly identified by **SOMCOT** after sufficiently many samples.

**Model selection.** Another important use case is identifying the hidden dynamics underlying realizations of stochastic processes. We model this scenario in two experiments. In the first one, we generate a block-structured Markov chain $M_{\mathcal{Y}}^*$, and a set of low-dimensional Markov chains $M_{\mathcal{X}}$ with different transition kernels parameterized by $\theta \in [0, 1]$. This set contains the true model $M_{\mathcal{X}}^*$ underlying $M_{\mathcal{Y}}^*$, corresponding to $\theta = 0.5$. We compute estimates of the distances between $M_{\mathcal{Y}}$ and all the candidates of the model class by running **SOMCOT** for various sample sizes, and show the results on Figure 2a. Notably, the distance achieves its minimum for the true model, and increases as $\theta$ is further separated from its true value.

We also conduct a second experiment on model selection in *continuous* state spaces. To this end, we consider the classic control environment *Pendulum-v1* from Gymnasium [Towers et al., 2024]. We begin by training a near-optimal policy using the DDPG algorithm [Lillicrap et al., 2015] and fix this policy to induce a Markov chain over the environment. The continuous state variables of the pendulum are then discretized to $n$ bins each. We instantiate several copies of the environment by varying a hyperparameter: the acceleration constant $g$, which by default is set to $g = 9.8$. The learning task we consider is to identify which of the class of candidate models best explains the unknown true dynamics corresponding to the default choice of $g$. Figure 2b shows the results obtained. Again, we can observe that the distance achieves its minimum for the true model, and increases as $g$ departs from its true value. Notably, due to discretization of the state space, the observations are not Markovian, yet the results clearly indicate that **SOMCOT** is still able to produce meaningful distance estimates, thus illustrating the potential of this methodology for general representation learning tasks.

## 6 Discussion

In this work, we have explored the use of stochastic methods to compute distances between Markov chains. This is still a largely unexplored field, and we believe the results presented here open the door to many interesting advances. We outline some of these future research directions we consider to be the most promising.

Most importantly, it remains unclear how to properly scale our algorithm to larger problems with potentially infinite state spaces. While we believe that our bounds cannot be improved significantly in the case of finite state spaces, addressing infinite state spaces should be possible under appropriate structural assumptions. One may take direct inspiration from the OT literature to extend our approach to these settings. For instance, parametrizing the dual variables via kernels or neural networks has been shown to be an effective approach to solve static OT problems (cf. Genevay et al. 2016, Seguy et al. 2018), and extending this idea to our setting is straightforward. The real challenge seems to be approximating the primal variables, which correspond to (conditional) probability distributions, which are not straightforward to parametrize via modern architectures (at least as long as one is interested in theoretically sound methods). We leave the investigation of this very interesting question open for future work.

Among all applications of optimal transport for Markov chains, its use in representation learning for RL is particularly interesting to us. Many previous works on this domain have highlighted the potential of bisimulation metrics for learning state abstractions, but all theoretically sound previous methods for computing such distances required full knowledge of the transition kernels. Removing this need brings us closer to realizing this potential. The experiments presented here demonstrate the effectiveness of bisimulation metrics in capturing symmetries and latent dynamics of Markov chains directly from sampled trajectories, both in random walks and discretized classical control environments. Incorporating function approximation along the lines mentioned above could significantly enhance these applications.

Besides the already-mentioned interpretation of the variables $\lambda_x$ and $\lambda_y$ as encoder-decoder maps, there are other side products of **SOMCOT** that can prove useful for representation learning. Most notably, the optimal dual variables $\alpha_{\mathcal{X}}$ and $\alpha_{\mathcal{Y}}$ correspond to the derivatives of the distance with respect to the state-transition distributions $\nu_{\mathcal{X}}$ and $\nu_{\mathcal{Y}}$, which is a fact that can prove extremely useful for the development of practical methods. Indeed, notice that these distributions themselves are differentiable with respect to the transition kernels, which altogether allows one to backpropagate through the OT distance as a loss function in representation learning tasks. Successful implementation of this idea may lead to strong theoretically sound alternatives to empirically successful methods such as MuZero [Schrittwieser et al., 2020]. This latter method uses a loss function remarkably similar to our OT distance, albeit with some limitations that disallow its application to stochastic environments (cf. Jiang 2024). Once again, we leave this direction for future work.

**Acknowledgements.** The authors wish to thank Csaba Szepesvári for thought-provoking discussions during the preparation of this manuscript. This project has received funding from the European Research Council (ERC), under the European Union's Horizon 2020 research and innovation programme (Grant agreement No. 950180). This work has been co-funded by MICIU/AEI/UE-PID2023-147145NB-I00, AGAUR SGR and MCIN/AEI/10.13039/501100011033 under the Maria de Maeztu Units of Excellence Programme (CEX2021-001195-M).

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

# A  Equivalence of the LP formulation and the bisimulation metric

In this section we prove Proposition 1, which together with Equation (2) implies that the novel LP formulation is equivalent to Equation (1) for computing the bisimulation metric. Our proof uses the linear programming formulation of Calo et al. [2024] as a starting point. Concretely, Calo et al. [2024] prove that $\mu$ is the induced occupancy coupling of a bicausal coupling $\pi \in \Pi_{\mathrm{bc}}$ if and only if $\mu$ satisfies the following set of constraints:

$$\sum_{x',y'} \mu(x,y,x',y') = \gamma \sum_{x',y'} \mu(x',y',x,y) + (1-\gamma)\nu_0(x,y) \quad (\forall x,y \in \mathcal{X}\times\mathcal{Y}), \qquad (23)$$

$$\sum_{y'} \mu(x,y,x',y') = \sum_{x'',y'} \mu(x,y,x'',y')P_{\mathcal{X}}(x'|x) \qquad (\forall x,y,x' \in \mathcal{X}\times\mathcal{Y}\times\mathcal{X}), \qquad (24)$$

$$\sum_{x'} \mu(x,y,x',y') = \sum_{x',y''} \mu(x,y,x',y'')P_{\mathcal{Y}}(y'|y) \qquad (\forall x,y,y' \in \mathcal{X}\times\mathcal{Y}\times\mathcal{Y}). \qquad (25)$$

Importantly, the above constraints provide a complete characterization of occupancy couplings: not only do occupancy couplings satisfy all equations, but *any* $\mu$ satisfying the three linear systems of equations above is a valid occupancy coupling (cf. Lemma 1 in Calo et al., 2024).

To prove the proposition it is sufficient to show that $\mu$ satisfies Equations (6)–(8) if and only if it satisfies Equations (23)–(25). Equation (6) is identical to Equation (23), and thus we are left with showing that Equations (7) and (8) are equivalent to Equations (24) and (25), respectively. We will show the first of these claims, and note that the second claim will follow by symmetry. Within the proof, we will repeatedly make use of the shorthand notation $\nu_{\mathcal{X}}(x) = \sum_{x'} \nu_{\mathcal{X}}(x,x')$ and the easy-to-see fact that $P_{\mathcal{X}}(x'|x) = \nu_{\mathcal{X}}(x,x')/\nu_{\mathcal{X}}(x)$.

First, let us assume that $\mu \in \mathbb{R}^{\mathcal{X}\times\mathcal{Y}\times\mathcal{X}\times\mathcal{Y}}$ satisfies Equation (24). Then, it is easy to check that Equation (7) is satisfied with the choice $\lambda_{\mathcal{X}}(y|x) = \sum_{x',y'} \mu(x,y,x',y')/\nu_{\mathcal{X}}(x)$, making use of the relation between $\nu_{\mathcal{X}}$ and $P_{\mathcal{X}}$ stated above. Conversely, assume that $\mu, \lambda_{\mathcal{X}}$ satisfy Equation (7). Summing both sides over $x'$ gives $\sum_{x',y'} \mu(x,y,x',y') = \nu_{\mathcal{X}}(x)\lambda_{\mathcal{X}}(y|x)$, which can be plugged back into Equation (7) to obtain

$$\sum_{y'} \mu(x,y,x',y') = \nu_{\mathcal{X}}(x,x')\lambda_{\mathcal{X}}(y|x) = P_{\mathcal{X}}(x'|x)\nu_{\mathcal{X}}(x)\lambda_{\mathcal{X}}(y|x) = P_{\mathcal{X}}(x'|x)\sum_{x',y'} \mu(x,y,x'',y'),$$

thus confirming that Equation (24) is indeed satisfied.

For the last part, let us define $\mu_{\mathcal{X}}$ as $\mu_{\mathcal{X}}(x,x') = \sum_{y,y'} \mu(x,y,x',y')$ for each $(x,x')$ whenever Equations (6)–(8) are satisfied. As per the above argument, Equations (3) and (24) are also satisfied, and thus summing both equations over $y$ yields

$$\sum_{x'} \mu_{\mathcal{X}}(x,x') = \gamma \sum_{x'} \mu_{\mathcal{X}}(x',x) + (1-\gamma)\nu_{0,\mathcal{X}}(x),$$

$$\mu_{\mathcal{X}}(x,x') = P_{\mathcal{X}}(x'|x)\sum_{x''} \mu_{\mathcal{X}}(x,x'').$$

A standard argument (provided as Lemma 12 in Appendix E) shows that the unique solution to this system of equations is equal to the marginal occupancy measure $\nu_{\mathcal{X}}$. This implies

$$\sum_{y} \lambda_{\mathcal{X}}(y|x) = \sum_{y} \frac{\sum_{x',y'} \mu(x,y,x',y')}{\nu_{\mathcal{X}}(x)} = \frac{\sum_{x'} \mu_{\mathcal{X}}(x,x')}{\nu_{\mathcal{X}}(x)} = \frac{\sum_{x'} \nu_{\mathcal{X}}(x,x')}{\nu_{\mathcal{X}}(x)} = 1,$$

which concludes the proof.

# B  Further details about the algorithm

In this section we describe some further details about the derivation of our algorithm (**SOMCOT**) that were omitted from the main text. Algorithm 2 provides a full pseudocode for **SOMCOT**.

At a high level, the algorithm aims to find the saddle point of the Lagrangian (9) by performing primal-dual updates for the two sets of variables $(\mu, \lambda)$ and $(\alpha, V)$, referred to as *minimizing* and

*maximizing players*, respectively (or often simply call them min and max players). Both sets of players maintain a sequence of iterates $(\mu_k, \lambda_k)$ and $(\alpha_k, V_k)$, which are updated using versions of online stochastic mirror descent, described below in detail. For the updates, the $\mu$ and $\lambda$ players move in the direction of the negative gradient of the Lagrangian evaluated at $(\mu_k, \lambda_k; \alpha_k, V_k)$, and the $\alpha$ and $V$ players move in the direction of the positive gradient.

Since some of these gradients involve the occupancy measures $\nu_{\mathcal{X}}$ and $\nu_{\mathcal{Y}}$, they cannot be computed exactly without perfect knowledge of these distributions. However, since the dependence on $\nu_{\mathcal{X}}$ and $\nu_{\mathcal{Y}}$ is always linear, it is straightforward to obtain unbiased gradient estimators given only sample access to the chains $M_{\mathcal{X}}$ and $M_{\mathcal{Y}}$. We provide a detailed guide for sampling from these distributions in Appendix B.4.

The remainder of the section provides a detailed derivation of the gradients and update rules used in each iteration. We begin by introducing the Mirror Descent algorithm that forms the basis of the update rules for each variable.

---

**Algorithm 2** Stochastic Optimization for Markov Chain Optimal Transport (**SOMCOT**)

---

**Require:** Convex sets $\mathcal{D}_{\alpha_{\mathcal{X}}} \subset \mathbb{R}^{\mathcal{X} \times \mathcal{X} \times \mathcal{Y}}, \mathcal{D}_{\alpha_{\mathcal{Y}}} \subset \mathbb{R}^{\mathcal{X} \times \mathcal{Y} \times \mathcal{Y}}, \mathcal{D}_V \subset \mathbb{R}^{\mathcal{X} \times \mathcal{Y}}$,
    Initial values $\mu_1, \lambda_{\mathcal{X},1}, \lambda_{\mathcal{Y},1}, \alpha_{\mathcal{X},1}, \alpha_{\mathcal{Y},1}, V_1$,
    Learning rates $\eta, \eta_{\mathcal{X}}, \eta_{\mathcal{Y}}, \beta_{\mathcal{X}}, \beta_{\mathcal{Y}}, \beta > 0$.
1: **for** $k = 1, \ldots K - 1 :$ **do**
2:     **Step 1: Draw samples from the Markov chains**
3:     Receive $(X_k, X_k') \sim \nu_{\mathcal{X}}, (Y_k, Y_k') \sim \nu_{\mathcal{Y}}$
4:     **Step 2: Compute gradients or stochastic gradients**
5:     $g_{k,\mu}(x, y, x', y') \leftarrow c(x, y) - \alpha_{\mathcal{X},k}(x, x', y) - \alpha_{\mathcal{Y},k}(x, y, y') + \gamma V_k(x', y') - V_k(x, y)$
6:     $\widetilde{g}_{k,\lambda_{\mathcal{X}}}(y|x) \leftarrow \mathbf{1}_{\{X_k = x\}} \alpha_{\mathcal{X},k}(x, X_k', y)$
7:     $\widetilde{g}_{k,\lambda_{\mathcal{Y}}}(x|y) \leftarrow \mathbf{1}_{\{Y_k = y\}} \alpha_{\mathcal{Y},k}(x, y, Y_k')$
8:     $\widetilde{g}_{k,\alpha_{\mathcal{X}}}(x, x', y) \leftarrow \sum_{y'} \mu_k(x, y, x', y') - \mathbf{1}_{\{X_k, X_k' = x, x'\}} \lambda_{\mathcal{X},k}(y|x)$
9:     $\widetilde{g}_{k,\alpha_{\mathcal{Y}}}(x, y, y') \leftarrow \sum_{x'} \mu_k(x, y, x', y') - \mathbf{1}_{\{Y_k, Y_k' = y, y'\}} \lambda_{\mathcal{Y},k}(x|y)$
10:    $g_{k,V}(x, y) \leftarrow \sum_{x',y'} \mu_k(x, y, x', y') - (1 - \gamma)\nu_0(xy) - \gamma \sum_{\widehat{x},\widehat{y}} \mu_k(\widehat{x}, \widehat{y}, x, y)$
11:    **Step 3: Update primal variables**
12:    $\mu_{k+1}(x, y, x', y') \propto \mu_k(x, y, x', y') \exp(-\eta g_{k,\mu}(x, y, x', y'))$
13:    $\lambda_{\mathcal{X},k+1}(y|x) \propto \lambda_{\mathcal{X},k}(y|x) \exp(-\eta_{\mathcal{X}} \widetilde{g}_{k,\lambda_{\mathcal{X}}}(y|x))$
14:    $\lambda_{\mathcal{Y},k+1}(x|y) \propto \lambda_{\mathcal{Y},k}(x|y) \exp(-\eta_{\mathcal{Y}} \widetilde{g}_{k,\lambda_{\mathcal{Y}}}(x|y))$
15:    **Step 4: Update dual variables**
16:    $\alpha_{\mathcal{X},k+1} \leftarrow \Pi_{\mathcal{D}_{\alpha_{\mathcal{X}}}} (\alpha_{\mathcal{X},k} - \beta_{\mathcal{X}} \widetilde{g}_{k,\alpha_{\mathcal{X}}})$
17:    $\alpha_{\mathcal{Y},k+1} \leftarrow \Pi_{\mathcal{D}_{\alpha_{\mathcal{Y}}}} (\alpha_{\mathcal{Y},k} - \beta_{\mathcal{Y}} \widetilde{g}_{k,\alpha_{\mathcal{Y}}})$
18:    $V_{k+1} \leftarrow \Pi_{\mathcal{D}_V} (V_k - \beta g_{k,V})$
19: **end for**
20: Output $\overline{\mu}_K = \frac{1}{K} \sum_{k=1}^{K} \mu_k$.

---

### B.1 Online Stochastic Mirror Descent

Online Stochastic Mirror Descent (OSMD) is an algorithm for the problem of online linear optimization, where in a sequence of rounds $k = 1, 2, \ldots, K$, the following steps are repeated:

    1. The online learner picks a decision $z_k$ taking values in the vector space $\mathcal{Z}$,

    2. the environment picks a linear function $g_k : \mathcal{Z} \to \mathbb{R}$,

    3. the online learner incurs loss $\langle g_k, z_k \rangle$,

    4. the online learner observes an unbiased estimate $\widetilde{g}_k \in \mathcal{Z}^*$ of the loss function.

The sequence of steps above defines a filtration $(\mathcal{F}_k)_k$, and the loss estimate $\widetilde{g}_k$ is assumed to satisfy $\mathbb{E}[\widetilde{g}_k | \mathcal{F}_{k-1}] = g_k$. Typically, the vectors $g_k$ are subgradients of a sequence of convex loss functions, and thus we will often refer to them with this term, and also call the vectors $\widetilde{g}_k$ stochastic subgradients (or simply stochastic gradients). OSMD computes a sequence of updates based on these noisy gradient estimates and a convex and differentiable *distance-generating function* $\Psi : \mathcal{Z} \to \mathbb{R}$. Concretely,

OSMD operates with the Bregman divergence $\mathcal{B}_\Psi$ of $\Psi$, defined for each pair $z, z' \in \mathcal{Z}$ as

$$\mathcal{B}_\Psi(z\|z') = \Psi(z) - \Psi(z') - \langle \nabla \Psi(z'), z - z' \rangle.$$

OSMD starts with an initial point $z_1 \in \mathcal{Z}$, and computes each subsequent iterate using the recursive update rule

$$z_{k+1} = \underset{z \in \mathcal{Z}}{\arg\min} \ \langle \widetilde{g}_k, z \rangle + \frac{1}{\eta} \mathcal{B}_\Psi(z\|z_k), \tag{26}$$

where $\eta > 0$ is called the learning rate.

Each of the update rules used by **SOMCOT** follows from instantiating OSMD with a specific decision space $\mathcal{Z}$, a distance-generating function $\Psi$ and a noisy subgradient estimator. Concretely, we will make use of the following instances and corresponding update rules of MD, whose derivations are available in standard textbooks (e.g. Orabona 2019).

**Proposition 2.** *When $\mathcal{Z} = \mathbb{R}^d$ and $\Psi$ is the squared Euclidean norm defined as $\Psi(z) = \frac{1}{2}\|z\|_2^2$ for each $z \in \mathcal{Z}$, the OSMD update reduces to the* projected stochastic gradient descent *update rule*

$$z_{k+1} = \Pi_\mathcal{Z}(z_k - \eta \widetilde{g}_k),$$

*where $\Pi_\mathcal{Z}$ is the orthogonal projection onto the set $\mathcal{Z}$ defined as $\Pi_\mathcal{Z}(x) = \arg\min_{y \in \mathcal{Z}} \|x - y\|_2$.*

**Proposition 3.** *When $\mathcal{Z} = \Delta_\mathcal{X}$ is the probability simplex on a finite set $\mathcal{X}$ and $\Psi$ is the negative entropy defined as $\Psi(p) = \sum_x p(x) \log p(x)$, $p \in \Delta_\mathcal{X}$, the OSMD update reduces to*

$$p_{k+1}(x) = \frac{p_k(x)e^{-\eta \widetilde{g}_k(x)}}{\sum_y p_k(y)e^{-\eta \widetilde{g}_k(y)}} \quad (\forall x \in \mathcal{X}).$$

**Proposition 4.** *When $\mathcal{Z} = \Delta_{\mathcal{Y}|\mathcal{X}}$ is the conditional simplex on finite sets $\mathcal{X}$ and $\mathcal{Y}$ and $\Psi$ is the total negative entropy $\Psi(p) = \sum_{x,y} p(y|x) \log p(y|x)$, $p \in \Delta_{\mathcal{Y}|\mathcal{X}}$, the OSMD update reduces to*

$$p_{k+1}(y|x) = \frac{p_k(y|x)e^{-\eta \widetilde{g}_k(y|x)}}{\sum_{\bar{y}} p_k(\bar{y}|x)e^{-\eta \widetilde{g}_k(\bar{y}|x)}} \quad (\forall x, y \in \mathcal{X}\mathcal{Y}).$$

## B.2 Primal updates

In this section we derive the gradients and update rules of the primal variables $\mu$ and $\lambda_\mathcal{X}$. The gradient and update rule of $\lambda_\mathcal{Y}$ follow by symmetry.

For $\mu$, first notice that any valid occupancy coupling $\mu$ is an element of the simplex $\Delta_{\mathcal{X}\mathcal{Y}\mathcal{X}\mathcal{Y}}$: summing the flow constraint in (6) over $x$ and $y$ immediately yields $\sum_{x,y,x',y'} \mu(x, y, x', y') = 1$. Thus, it is natural to enforce this constraint throughout the execution of the algorithm and use OSMD with the entropy regularizer given in Proposition 3. In order to derive the update rule, it remains to compute the gradients of the Lagrangian with respect to $\mu$, which is given as

$$\frac{\partial \mathcal{L}}{\partial \mu}[\mu, \lambda; \alpha, V](x, y, x', y') = c(x, y) - \alpha_\mathcal{X}(x, x', y) - \alpha_\mathcal{Y}(x, y, y') + \gamma V(x', y') - V(x, y). \tag{27}$$

In each iteration $k$, the algorithm computes the gradient $g_{k,\mu} = \frac{\partial \mathcal{L}}{\partial \mu}[\mu_k, \lambda_{\mathcal{X},k}, \lambda_{\mathcal{Y},k}; \alpha_{\mathcal{X},k}, \alpha_{\mathcal{Y},k}, V_k]$, which can be used as the unbiased estimator $\widetilde{g}_k$. Altogether, this yields the update rule on line 12 of Algorithm 2.

As for the $\lambda_\mathcal{X}$ variables, notice that Proposition 1 implies that $\lambda_\mathcal{X}$ belongs to the conditional simplex $\Delta_{\mathcal{Y}|\mathcal{X}} = \{\lambda \in \mathbb{R}_+^{\mathcal{Y} \times \mathcal{X}}, \forall x \in \mathcal{X}, \sum_y \lambda(y|x) = 1\}$. Thus, it is natural to use the total negative entropy as regularization function (as suggested in Proposition 4). For selecting the update direction, we note that the gradient of the Lagrangian with respect to $\lambda_\mathcal{X}$ is

$$\frac{\partial \mathcal{L}}{\partial \lambda_\mathcal{X}}[\mu, \lambda; \alpha, V](y|x) = \sum_{x'} \nu_\mathcal{X}(x, x') \alpha_\mathcal{X}(x, x', y) = \mathbb{E}_{X, X' \sim \nu_\mathcal{X}} \left[ \mathbf{1}_{\{X=x\}} \alpha_\mathcal{X}(x, X', y) \right]. \tag{28}$$

Thus, we can obtain a stochastic gradient estimate $\widetilde{g}_{k,\lambda_\mathcal{X}}$ of $\frac{\partial \mathcal{L}}{\partial \lambda_\mathcal{X}}[\mu_k, \lambda_{\mathcal{X},k}, \lambda_{\mathcal{Y},k}; \alpha_{\mathcal{X},k}, \alpha_{\mathcal{Y},k}, V_k]$ by sampling a transition $(X_k, X_k')$ from $\nu_\mathcal{X}$ and setting $\widetilde{g}_{k,\lambda_\mathcal{X}} = \mathbf{1}_{\{X_k=x\}} \alpha_{\mathcal{X},k}(x, X_k', y)$. Putting things together, this yields the update rules on lines 13–14 of Algorithm 2.

## B.3 Dual updates

We now move our attention to the dual variables. Again, we will derive the gradients and update rules for $\alpha_\mathcal{X}$ and $V$, and the gradient and update rule for $\alpha_\mathcal{Y}$ follow by symmetry. Since we are maximizing over the dual variables which are not restricted to any simplex, we update them using projected (stochastic) gradient ascent, which is why the gradients are negated below. The feasible sets for the dual variables are chosen to enable using Lemma 2 for bounding the estimation error—see Appendix C.1 for details.

The gradient of the Lagrangian with respect to $\alpha_\mathcal{X}$ is defined as

$$\frac{\partial \mathcal{L}}{\partial \alpha_\mathcal{X}}[\mu, \lambda; \alpha, V](x, x', y) = -\left(\sum_{y'} \mu(x, y, x', y') - \nu_\mathcal{X}(x, x')\lambda_\mathcal{X}(y|x)\right)$$

$$= -\mathbb{E}_{X_k, X_k' \sim \nu_\mathcal{X}}\left[\sum_{y'} \mu(x, y, x', y') - \mathbf{1}_{\{X_k, X_k' = x, x'\}}\lambda_\mathcal{X}(y|x)\right].$$

Our algorithm will use the update direction $\widetilde{g}_{k, \alpha_\mathcal{X}}(x, x', y) = \sum_{y'} \mu_k(x, y, x', y') - \mathbf{1}_{\{X_k, X_k' = x, x'\}}\lambda_{\mathcal{X}, k}(y|x)$. We will apply OSMD to update $\alpha_\mathcal{X}$ using a learning rate $\beta_\mathcal{X}$ and the regularizer in Proposition 2, which yields the update rule on lines 16–17 of Algorithm 2.

The gradient of the Lagrangian with respect to $V$ is given by

$$\frac{\partial \mathcal{L}}{\partial V}[\mu, \lambda; \alpha, V](xy) = -\left(\sum_{x', y'} \mu(x, y, x', y') - (1 - \gamma)\nu_0(x, y) - \gamma \sum_{\widehat{x}, \widehat{y}} \mu(\widehat{x}, \widehat{y}, x, y)\right).$$

We use the update direction $g_{k, V} = \sum_{x', y'} \mu_k(x, y, x', y') - (1 - \gamma)\nu_0(x, y) - \gamma \sum_{\widehat{x}, \widehat{y}} \mu_k(\widehat{x}, \widehat{y}, x, y)$ and apply OSMD to update $V$ using a learning rate $\beta$ and the regularizer in Proposition 2, which yields the update rule on line 18 of Algorithm 2.

## B.4 Sampling from $\nu_\mathcal{X}$ and $\nu_\mathcal{Y}$

A key step in constructing our gradient estimators (and thus running our algorithm) is drawing samples from the occupancy measures $\nu_\mathcal{X}$ and $\nu_\mathcal{Y}$. Here we provide further details about how to perform this operation in practice.

In order to generate a sample from the occupancy measure, we let $G$ be a geometric random variable with mean $\frac{1}{1-\gamma}$, and recall the definition of the marginal occupancy measure $\nu_\mathcal{X}$ to write

$$\nu_\mathcal{X}(x, x') = (1 - \gamma)\sum_{t=0}^{\infty} \gamma^t \mathbb{P}\left[X_t = x, X_{t+1} = x'\right] = \sum_{t=0}^{\infty} \mathbb{P}\left[G = t\right] \mathbb{P}\left[X_t = x, X_{t+1} = x'\right]$$

$$= \mathbb{P}\left[X_G = x, X_{G+1} = x'\right].$$

Thus, one can obtain independent samples from $\nu_\mathcal{X}$ by first sampling a geometric stopping time $G$, sample a sequence $(X_0, X_1, \ldots, X_G, X_{G+1})$, and keep the last pair of states $X_G, X_{G+1}$.

We remark that the task of sampling from an occupancy measure is common in reinforcement learning, and in particular it is necessary for correctly implementing policy gradient methods. To avoid sampling an entire sequence in each iteration, it is standard practice to replace samples from the occupancy measure with arbitrary sample trajectories generated by the Markov chain. Specifically, it is common to ignore discounting and draw samples directly from trajectories in which consecutive state pairs are no longer independent. We expect that, like most other RL algorithms, our method is also resilient to such abuse, and can be fed with sample pairs drawn from longer trajectories without resets or throwing away samples to ensure independence.

## C  Analysis

This section provides the complete details for the proof of our main result, Theorem 1. Throughout the analysis, we will assume $\|c\|_\infty \leq 1$. Completing the outline provided in Section 4 requires filling

two gaps: proving Lemma 1, and bounding the duality gap in terms of the regrets of the two players. These are respectively done in Sections C.1 and C.3 below (with Section C.2 providing additional technical tools for the proof of Lemma 1). Putting the two parts together complete the proof.

## C.1 Proof of Lemma 1

The majority of our theoretical analysis is dedicated to proving the error bound stated as Lemma 1, recalled here for convenience as

$$|\langle \overline{\mu}_K - \mu^*, c\rangle| \leq \mathcal{G}_K(\mu^*, \lambda^*, \alpha^*, V^*). \tag{29}$$

As a first step towards this proof, we first need to define a technical tool that will allow us to quantify the constraint violations associated with the output $\overline{\mu}_K$. Indeed, one challenge in the analysis is that $\overline{\mu}_K$ does not necessarily satisfy the constraints (6)–(8) exactly. We quantify this effect by defining *total absolute constraint violations* associated with the primal variables $\mu$, $\lambda_\mathcal{X}$ and $\lambda_\mathcal{Y}$ respectively by

$$\partial\mathcal{F}(\mu) = \sum_{x,y} \left| \sum_{x',y'} \mu(x,y,x'y') - \gamma \sum_{\widehat{x},\widehat{y}} \mu(\widehat{x},\widehat{y},x,y) - (1-\gamma)\nu_0(x,y) \right|$$

$$\partial\mathcal{C}_\mathcal{X}(\mu, \lambda_\mathcal{X}) = \sum_{x,x',y} \left| \sum_{y'} \mu(x,y,x',y') - \nu_\mathcal{X}(x,x')\lambda_\mathcal{X}(y|x) \right|$$

$$\partial\mathcal{C}_\mathcal{Y}(\mu, \lambda_\mathcal{Y}) = \sum_{x,y,y'} \left| \sum_{x'} \mu(x,y,x',y') - \nu_\mathcal{Y}(y,y')\lambda_\mathcal{Y}(x|y) \right|.$$

For the sake of analysis, we will make use of a *rounding procedure* that will convert $\overline{\mu}_K$ into a valid occupancy coupling $r(\overline{\mu}_K)$ that satisfies all constraints. Importantly, this rounding procedure never has to be executed in reality: it is only used as a device within the analysis. The details of this rounding process (which is an adaptation of a method developed by Calo et al. 2024) are provided in Appendix C.2. The following lemma provides an upper bound on the rounding error in terms of the total absolute constraint violations.

**Lemma 2.** *Let $\mu \in \Delta_{\mathcal{X}\mathcal{Y}\mathcal{X}\mathcal{Y}}$ and $r(\mu)$ be its rounding (as defined in Appendix C.2), and $\lambda_\mathcal{X}$ and $\lambda_\mathcal{Y}$ be arbitrary. Then, we have*

$$\|\mu - r(\mu)\|_1 \leq \frac{3\mathcal{C}_\mathcal{X}(\mu, \lambda_\mathcal{X}) + 3\mathcal{C}_\mathcal{Y}(\mu, \lambda_\mathcal{Y}) + \partial\mathcal{F}(\mu)}{1-\gamma}. \tag{30}$$

The proof is provided along with all relevant definitions in Appendix C.2. With this rounding process and its guarantees at hand, we can rewrite the absolute error between the cost estimate $\langle \overline{\mu}_K, c\rangle$ and the true cost $d(M_\mathcal{X}, M_\mathcal{Y}) = \langle \mu^*, c\rangle$ as follows:

$$\begin{aligned} |\langle \overline{\mu}_K - \mu^*, c\rangle| &\leq \langle r(\overline{\mu}_K) - \mu^*, c\rangle + \|r(\overline{\mu}_K) - \overline{\mu}_K\|_1 \|c\|_\infty \\ &\leq \langle \overline{\mu}_K - \mu^*, c\rangle + 2\|\overline{\mu}_K - r(\overline{\mu}_K)\|_1 \|c\|_\infty. \end{aligned} \tag{31}$$

Here, the first step follows from the triangle inequality and the crucially important fact that $\langle r(\overline{\mu}_K) - \mu^*, c\rangle \geq 0$ thanks to the feasibility of $r(\overline{\mu}_K)$ and the optimality of $\mu^*$.

It now only remains to relate the quantity appearing on the right-hand side of the above bound with the duality gap. To this end, we define the shorthand $\overline{\lambda}_\mathcal{X} = \frac{1}{K}\sum_{k=1}^K \lambda_{\mathcal{X},k}$ and $\overline{\lambda}_\mathcal{Y} = \frac{1}{K}\sum_{k=1}^K \lambda_{\mathcal{Y},k}$ and recall the choice

$$(\alpha^*, V^*) = \underset{\alpha\in\mathcal{D}_\alpha, V\in\mathcal{D}_V}{\arg\max} \frac{1}{K}\sum_{k=1}^K \mathcal{L}(\mu_k, \lambda_k; \alpha, V).$$

Then, by plugging these variables into the Lagrangian, it is easy to check that

$$\frac{1}{K}\sum_{k=1}^K \mathcal{L}(\mu_k, \lambda_k; \alpha^*, V^*) = \langle \overline{\mu}_K, c\rangle + \frac{6\mathcal{C}_\mathcal{X}(\overline{\mu}_K, \overline{\lambda}_\mathcal{X}) + 6\mathcal{C}_\mathcal{Y}(\overline{\mu}_K, \overline{\lambda}_\mathcal{Y}) + 2\partial\mathcal{F}(\overline{\mu}_K)}{1-\gamma},$$

which, by using Lemma 2, implies the following bound:

$$\langle \overline{\mu}_K, c \rangle + 2 \left\| \overline{\mu}_K - r(\overline{\mu}_K) \right\|_1 \leq \frac{1}{K} \sum_{k=1}^{K} \mathcal{L}(\mu_k, \lambda_k; \alpha^*, V^*)$$

On the other hand, it is easily verified that $\langle \mu^*, c \rangle = \mathcal{L}(\mu^*, \lambda^*; \alpha, V)$ holds for any choice of $\alpha$ and $V$, thanks to the fact that $\mu^*$ and $\lambda^*$ verify all the constraints of the LP. Putting this together with Equation (31), we obtain that the error can be bounded in terms of the duality gap at the above-defined comparator $(\mu^*, \lambda^*, \alpha^*, V^*)$ as

$$|\langle \overline{\mu}_K - \mu^*, c \rangle| \leq \mathcal{G}_K(\mu^*, \lambda^*, \alpha^*, V^*).$$

This concludes the proof of Lemma 1.

## C.2  Rounded coupling and rounding error

We describe the process and guarantees of rounding an (approximate) occupancy coupling $\mu \in \mathbb{R}_+^{\mathcal{X}\mathcal{Y}\mathcal{X}\mathcal{Y}}$. We note that computing this rounding requires knowledge of $\nu_X$, but this does not cause any practical problems since the rounding is only ever executed in the analysis. For the rounding process itself, we first introduce the *state-occupancy measure* $\nu_\mu(x, y) = \sum_{x'y'} \mu(x, y, x', y')$, and we define the associated *transition coupling* $\pi_\mu$ as the kernel $\pi_\mu : \mathcal{X}\mathcal{Y} \to \Delta_{\mathcal{X}\mathcal{Y}}$ with entries

$$\pi_\mu(x', y'|x, y) = \begin{cases} \frac{\mu(x,y,x',y')}{\nu_\mu(x,y)} & \text{if } \nu_\mu(x, y) \neq 0, \\ P_{\mathcal{X}}(x'|x) P_{\mathcal{Y}}(y'|y) & \text{otherwise.} \end{cases}$$

As shown by Calo et al. [2024], each transition coupling $\pi : \mathcal{X}\mathcal{Y} \to \Delta_{\mathcal{X}\mathcal{Y}}$ induces a unique occupancy coupling $\mu^\pi$, and that the occupancy induced by $\pi_\mu$ is valid if and only if it equals $\mu$ (i.e., if $\mu^{\pi_\mu} = \mu$ holds). For more details, we refer to Appendix B.2 in Calo et al. [2024].

Following Calo et al. [2024], we will apply the rounding procedure of Altschuler et al. [2017, Algorithm 2] to $\pi_\mu$ to obtain a valid transition coupling $r(\pi_\mu)$, and then extract the occupancy coupling induced by $r(\pi_\mu)$. More precisely, for two probability distributions $p \in \Delta(\mathcal{X}), q \in \Delta(\mathcal{Y})$, the set of valid couplings is defined as $\mathcal{U}_{p,q} = \{P \in \mathbb{R}_+^{\mathcal{X}\times\mathcal{Y}} : P \cdot \mathbf{1} = p; P^T \cdot \mathbf{1} = q\}$. For a nonnegative matrix $F \in \mathbb{R}_+^{\mathcal{X}\times\mathcal{Y}}$, the rounding procedure outputs a valid coupling $r(F, p, q) \in \mathcal{U}_{p,q}$. By Lemma 7 of Altschuler et al. [2017], the rounded coupling satisfies

$$\|r(F, p, q) - F\|_1 \leq 2(\|F \cdot \mathbf{1}\| + \|F^T \cdot \mathbf{1}\|).$$

For completeness the procedure is detailed in Algorithm 3.

---
**Algorithm 3** Rounding procedure for couplings

---
**Input:**  approximate coupling $F$, marginals $p$, $q$
$X \leftarrow \operatorname{diag}(\min(p/(F \cdot \mathbf{1}), \mathbf{1}))$
$F' \leftarrow XF$
$Y \leftarrow \operatorname{diag}(\min(q/(F'^\top \cdot \mathbf{1}), \mathbf{1}))$
$F'' \leftarrow F'Y$
$\operatorname{err}_p = p - F'' \cdot \mathbf{1}, \operatorname{err}_q = q - F''^\top \cdot \mathbf{1}$
**Output:**  $G \leftarrow F'' + \operatorname{err}_p \operatorname{err}_q^\top / \|\operatorname{err}_p\|_1$

---

This procedure is not symmetric, and thus we consider the following symmetrized procedure defined as

$$r^{\text{sym}}(F, p, q) = \frac{r(F, p, q) + r(F^T, q, p)^T}{2}.$$

To obtain the rounded transition coupling, we apply the rounding procedure individually for each pair of states $x, y$. In particular, for a transition kernel $\pi_\mu : \mathcal{X}\mathcal{Y} \to \Delta_{\mathcal{X}\mathcal{Y}}$, we define its rounded counterpart $\widetilde{\pi} = r(\pi)$ with entries

$$\widetilde{\pi}(\cdot|x, y) = r^{\text{sym}}(\pi(\cdot|x, y), P_{\mathcal{X}}(\cdot|x), P_{\mathcal{Y}}(\cdot|y)).$$

With some abuse of notation, we will now denote as $r(\mu)$ the occupancy coupling induced by $r(\pi_\mu)$. Because $r(\pi_\mu)$ is a valid transition coupling, $r(\mu)$ is a valid occupancy coupling. The following

derivations will relate the distance between $r(\mu)$ and $\mu$ to the total absolute constraint violations of $\mu$, thus providing a proof for Lemma 2.

To make the subsequent derivations easier, we will define some handy notation. We first define the operator $E : \Delta_{\mathcal{XYXY}} \to \Delta_{\mathcal{XY}}$ via its action on any $\mu$ as $(E\mu)(x, y) = \sum_{\widehat{x}, \widehat{y}} \mu(\widehat{x}, \widehat{y}, x, y)$, and note that this allows us to rewrite the flow condition (6) in the form $\nu_\mu = \gamma E\mu + (1 - \gamma)\nu_0$. For a state-distribution $\nu \in \Delta_{\mathcal{XY}}$ and a kernel $\pi : \mathcal{XY} \to \Delta_{\mathcal{XY}}$, we define the composition $\nu \circ \pi$ as the distribution $p$ with entries $p(x, y, x', y') = \nu(x, y)\pi(x', y'|x, y)$. We will specifically use the notation $\Delta(\mu) = \nu_\mu \circ (r(\pi_\mu) - \pi_\mu)$. Armed with all this notation, we bound the $\ell_1$ distance between $r(\mu)$ and $\mu$ as

$$
\begin{aligned}
\|r(\mu) - \mu\|_1 &= \left\| \nu_{r(\mu)} \circ r(\pi_\mu) - \nu_\mu \circ \pi_\mu \right\| \\
&= \left\| \nu_{r(\mu)} \circ r(\pi_\mu) - \nu_\mu \circ r(\pi_\mu) + \nu_\mu \circ r(\pi_\mu) - \nu_\mu \circ \pi_\mu \right\| \\
&\leq \left\| \nu_{r(\mu)} - \nu_\mu \right\|_1 + \left\| \nu_\mu \circ (r(\pi_\mu) - \pi_\mu) \right\|_1 \\
&= \left\| \nu_{r(\mu)} - (1 - \gamma)\nu_0 + (1 - \gamma)\nu_0 - \nu_\mu \right\|_1 + \|\Delta(\mu)\|_1 \\
&= \left\| \gamma E r(\mu) - \gamma E\mu + [\gamma E\mu + (1 - \gamma)\nu_0 - \nu_\mu] \right\|_1 + \|\Delta(\mu)\|_1 \\
&\leq \gamma \|r(\mu) - \mu\|_1 + \partial\mathcal{F}(\mu) + \|\Delta(\mu)\|_1 \,,
\end{aligned}
$$

where the second-to-last line uses the fact that $r(\mu)$ is a valid occupancy coupling and as such satisfy the flow condition (6), and we have recalled the definition of $\partial\mathcal{F}(\mu)$ stated in the main text. After reordering, we obtain

$$
\|r(\mu) - \mu\|_1 \leq \frac{\partial\mathcal{F}(\mu) + \|\Delta(\mu)\|_1}{1 - \gamma},
$$

and thus it remains to upper bound $\|\Delta(\mu)\|_1$. This is done in the following lemma, using which concludes the proof of Lemma 2.

**Lemma 3.** *For any $\mu \in \Delta_{\mathcal{XYXY}}$, and any $\lambda_\mathcal{X} : \mathcal{X} \to \Delta_\mathcal{Y}$ and $\lambda_\mathcal{Y} : \mathcal{Y} \to \Delta_\mathcal{X}$, we have $\|\Delta(\mu)\|_1 \leq 3\partial\mathcal{C}_\mathcal{Y}(\mu, \lambda_\mathcal{Y}) + 3\partial\mathcal{C}_\mathcal{X}(\mu, \lambda_\mathcal{X})$.*

*Proof.* By Lemma 7 of Altschuler et al. [2017], we have that for arbitrary state pairs $x, y$, the following is satisfied:

$$
\|r(\pi_\mu)(\cdot|x, y) - \pi_\mu(\cdot|x, y)\|_1
$$
$$
\leq 2 \left[ \sum_{x'} \left| P_\mathcal{X}(x'|x) - \sum_{y'} \pi_\mu(x', y'|x, y) \right| + \sum_{y'} \left| P_\mathcal{Y}(y'|y) - \sum_{x'} \pi_\mu(x', y'|x, y) \right| \right].
$$

By symmetry, this directly gives

$$
\|r^{\text{sym}}(\pi_\mu)(\cdot|x, y) - \pi_\mu(\cdot|x, y)\|_1
$$
$$
\leq \frac{3}{2} \left[ \sum_{x'} \left| P_\mathcal{X}(x'|x) - \sum_{y'} \pi_\mu(x', y'|x, y) \right| + \sum_{y'} \left| P_\mathcal{Y}(y'|y) - \sum_{x'} \pi_\mu(x', y'|x, y) \right| \right].
$$

Now, multiplying both sides by $\nu_\mu(x, y)$, we get

$$
\Delta(\mu)(x, y) = \nu_\mu(x, y) \|r^{\text{sym}}(\pi_\mu)(\cdot|x, y) - \pi_\mu(\cdot|x, y)\|_1
$$
$$
\leq \frac{3}{2} \left[ \sum_{x'} \left| P_\mathcal{X}(x'|x)\nu_\mu(x, y) - \sum_{y'} \mu(x, y, x', y') \right| + \sum_{y'} \left| P_\mathcal{Y}(y'|y)\nu_\mu(x, y) - \sum_{x'} \mu(x, y, x', y') \right| \right].
$$

The first term on the right-hand side of the above expression can be bounded as follows:

$$
\sum_{x'} \left| P_\mathcal{X}(x'|x)\nu_\mu(x, y) - \sum_{y'} \mu(x, y, x', y') \right|
$$
$$
= \sum_{x'} \left| P_\mathcal{X}(x'|x)\nu_\mu(x, y) - \nu_\mathcal{X}(x, x')\lambda_\mathcal{X}(y|x) + \nu_\mathcal{X}(x, x')\lambda_\mathcal{X}(y|x) - \sum_{y'} \mu(x, y, x', y') \right|
$$

$$\overset{(i)}{\leq} \sum_{x'} |P_{\mathcal{X}}(x'|x)\nu_\mu(x,y) - \nu_{\mathcal{X}}(x,x')\lambda_{\mathcal{X}}(y|x)| + \left|\nu_{\mathcal{X}}(x,x')\lambda_{\mathcal{X}}(y|x) - \sum_{y'}\mu(x,y,x',y')\right|$$

$$= \sum_{x'} P_{\mathcal{X}}(x'|x)|\nu_\mu(x,y) - \nu_{\mathcal{X}}(x)\lambda_{\mathcal{X}}(y|x)| + \partial\mathcal{C}_{\mathcal{X}}(\mu,\lambda_{\mathcal{X}})$$

$$\overset{(ii)}{=} |\nu_\mu(x,y) - \nu_{\mathcal{X}}(x)\lambda_{\mathcal{X}}(y|x)| + \partial\mathcal{C}_{\mathcal{X}}(\mu,\lambda_{\mathcal{X}})$$

$$\overset{(iii)}{=} \left|\sum_{x',y'}\mu(x,y,x',y') - \sum_{x'}\nu_{\mathcal{X}}(x)P_{\mathcal{X}}(x'|x)\lambda_{\mathcal{X}}(y|x)\right| + \partial\mathcal{C}_{\mathcal{X}}(\mu,\lambda_{\mathcal{X}})$$

$$\leq \sum_{x'}\left|\sum_{y'}\mu(x,y,x',y') - \nu_{\mathcal{X}}(x,x')\lambda_{\mathcal{X}}(y|x)\right| + \partial\mathcal{C}_{\mathcal{X}}(\mu,\lambda_{\mathcal{X}})$$

$$= 2\partial\mathcal{C}_{\mathcal{X}}(\mu,\lambda_{\mathcal{X}}).$$

Here, we used the triangle inequality for *(i)* and the fact that $\sum_{x'} P_{\mathcal{X}}(x'|x) = 1$ for *(ii)* and *(iii)*. The proof is concluded by repeating the same argument for the constraint violations $\partial\mathcal{C}_{\mathcal{Y}}(\mu,\lambda_{\mathcal{Y}})$, and plugging the results back into the previous inequalities. $\square$

### C.3  Regret bounds of the primal and dual sequence

This section provides an upper bound on the duality gap as defined in Equation (22), in terms of the *regrets* of the two set of algorithms controlling the primal and dual variables. Recalling the convention established in Appendix B, we will refer to the algorithms as the min- and max-players, with their regrets respectively defined as

$$\text{regret}_K^{\max}(\alpha^*, V^*) = \sum_{k=1}^{K}\big(\mathcal{L}(\mu_k,\lambda_k;\alpha^*,V^*) - \mathcal{L}(\mu_k,\lambda_k;\alpha_k,V_k)\big)$$

$$\text{regret}_K^{\min}(\mu^*,\lambda^*) = \sum_{k=1}^{K}\big(\mathcal{L}(\mu_k,\lambda_k;\alpha_k,V_k) - \mathcal{L}(\mu^*,\lambda^*;\alpha_k,V_k)\big),$$

where our notation emphasizes that each regret is measured against the comparators $\alpha^*, V^*$ and $\mu^*,\lambda^*$. With this notation, the duality gap can be rewritten as

$$\mathcal{G}_K(\mu^*,\lambda^*;\alpha^*,V^*) = \frac{\text{regret}_K^{\max}(\alpha^*,V^*) + \text{regret}_K^{\min}(\mu^*,\lambda^*)}{K}. \tag{32}$$

With a mild abuse of our earlier notation, we write out the full expression of the Lagrangian in terms of the $\alpha_{\mathcal{X}}, \alpha_{\mathcal{Y}}$ and $\lambda_{\mathcal{X}}, \lambda_{\mathcal{Y}}$ variables as $\mathcal{L}(\mu,\lambda_{\mathcal{X}},\lambda_{\mathcal{Y}};\alpha_{\mathcal{X}},\alpha_{\mathcal{Y}},V)$. The regret terms that need to be bounded can be further decomposed in terms of the following individual terms defined for each set of primal and dual variables:

$$\text{regret}_K^{\max}(\alpha_{\mathcal{X}}^*) = \sum_{k=1}^{K}\mathcal{L}(\mu_k,\lambda_{\mathcal{X},k},\lambda_{\mathcal{Y},k};\alpha_{\mathcal{X}}^*,\alpha_{\mathcal{Y}}^*,V^*) - \mathcal{L}(\mu_k,\lambda_{\mathcal{X},k},\lambda_{\mathcal{Y},k};\alpha_{\mathcal{X},k},\alpha_{\mathcal{Y}}^*,V^*)$$

$$\text{regret}_K^{\max}(\alpha_{\mathcal{Y}}^*) = \sum_{k=1}^{K}\mathcal{L}(\mu_k,\lambda_{\mathcal{X},k},\lambda_{\mathcal{Y},k};\alpha_{\mathcal{X},k},\alpha_{\mathcal{Y}}^*,V^*) - \mathcal{L}(\mu_k,\lambda_{\mathcal{X},k},\lambda_{\mathcal{Y},k};\alpha_{\mathcal{X},k},\alpha_{\mathcal{Y},k},V^*)$$

$$\text{regret}_K^{\max}(V^*) = \sum_{k=1}^{K}\mathcal{L}(\mu_k,\lambda_{\mathcal{X},k},\lambda_{\mathcal{Y},k};\alpha_{\mathcal{X},k},\alpha_{\mathcal{Y},k},V^*) - \mathcal{L}(\mu_k,\lambda_{\mathcal{X},k},\lambda_{\mathcal{Y},k};\alpha_{\mathcal{X},k},\alpha_{\mathcal{Y},k},V_k)$$

$$\text{regret}_K^{\min}(\mu^*) = \sum_{k=1}^{K}\mathcal{L}(\mu_k,\lambda_{\mathcal{X},k},\lambda_{\mathcal{Y},k};\alpha_{\mathcal{X},k},\alpha_{\mathcal{Y},k},V_k) - \mathcal{L}(\mu^*,\lambda_{\mathcal{X},k},\lambda_{\mathcal{Y},k};\alpha_{\mathcal{X},k},\alpha_{\mathcal{Y},k},V_k)$$

$$\text{regret}_K^{\min}(\lambda_{\mathcal{X}}^*) = \sum_{k=1}^{K}\mathcal{L}(\mu^*,\lambda_{\mathcal{X},k},\lambda_{\mathcal{Y},k};\alpha_{\mathcal{X},k},\alpha_{\mathcal{Y},k},V_k) - \mathcal{L}(\mu^*,\lambda_{\mathcal{X}}^*,\lambda_{\mathcal{Y},k};\alpha_{\mathcal{X},k},\alpha_{\mathcal{Y},k},V_k)$$

$$\text{regret}_K^{\min}(\lambda_{\mathcal{Y}}^*) = \sum_{k=1}^{K} \mathcal{L}(\mu^*, \lambda_{\mathcal{X}}^*, \lambda_{\mathcal{Y},k}; \alpha_{\mathcal{X},k}, \alpha_{\mathcal{Y},k}, V_k) - \mathcal{L}(\mu^*, \lambda_{\mathcal{X}}^*, \lambda_{\mathcal{Y}}^*; \alpha_{\mathcal{X},k}, \alpha_{\mathcal{Y},k}, V_k)$$

Thanks to the bilinearity of the Lagrangian, each of these terms can be seen as the regret of an online learning algorithm with linear loss / gain functions and decision variables taking values in a convex decision space $\mathcal{Z}$ (embedded within some Euclidean space). In particular, each of these regrets can be written in the following form for some sequences $(g_k)_k \in \mathbb{R}^d$, $(z_k)_k \in \mathcal{Z}$ and $z \in \mathcal{Z}$:

$$\text{regret}_K(z^*) = \sum_{k=1}^{K} \langle g_k, z_k - z^* \rangle.$$

As noted in Section B, our algorithm can be understood as running an instance of Online Stochastic Mirror Descent (OSMD) for each set of variables, and thus each regret term can be bounded using standard results. One challenge for the analysis is that the comparator points $\alpha^*$ and $V^*$ are chosen in a data-dependent manner. This is not easily handled by standard tools in online learning, but can still be treated with some relatively more advanced tools that are common in the context of saddle-point optimization (most notably, using techniques of Nemirovski et al. 2009, Rakhlin and Sridharan 2017). In particular, we will use the following general result to bound the regrets of each player in the analysis below.

**Lemma 4.** *Let $z^* \in \mathcal{Z}$ be a potentially data-dependent comparator and assume that $\Psi$ is $\lambda$-strongly convex with respect to some norm $\|\cdot\|$ whose dual is denoted by $\|\cdot\|_*$. Furthermore, suppose that $\sup_{z,z' \in \mathcal{Z}} \|z - z'\| \leq C$ holds for some constant $C > 0$. Then, for any $\widetilde{\eta} > 0$, the sequence $(z_k)_k$ produced by OSMD satisfies the following bound with probability at least $1 - \delta$:*

$$\sum_{k=1}^{K} \langle g_k, z_k - z^* \rangle \leq \frac{\mathcal{B}_\Psi(z\|z_1)}{\eta} + \frac{\eta}{2\lambda} \sum_{k=1}^{K} \|g_k\|_*^2$$

$$+ \frac{\mathcal{B}_\Psi(z\|z_1)}{\widetilde{\eta}} + \frac{\widetilde{\eta}}{2\lambda} \sum_{k=1}^{K} \|g_k - \widetilde{g}_k\|_*^2 + C \sqrt{2 \sum_{k=1}^{K} \|g_k - \widetilde{g}_k\|_*^2 \log \frac{1}{\delta}}.$$

While composed of standard elements, we provide the proof for the sake of completeness in Appendix D. The regret bound itself can be simplified in two different ways, depending on whether or not the algorithm in question uses deterministic or stochastic gradients: for deterministic updates, we have $g_k = \widetilde{g}_k$ and we can choose $1/\widetilde{\eta} = 0$, whereas for stochastic updates the choice $\widetilde{\eta} = \eta$ is more natural. This is how we will apply the lemma to each regret term below. In what follows, we will instantiate this bound to bound the regrets of all players listed above, which will require establishing *i)* the strong-convexity properties of the regularization functions, *ii)* bounds on the Bregman divergences between the initial points and the comparators and *iii)* bounds on the dual norms of the gradients and the gradient noise. This is done case by case in the following subsections.

### C.3.1 Regret of the $\alpha$-players

The policy of the $\alpha$-players is to run projected online stochastic gradient ascent on the feasible set $\mathcal{Z} = B_\infty\left(\frac{6}{1-\gamma}\right)$, and unbiased gradient estimators with elements defined respectively as

$$\widetilde{g}_{k,\alpha_{\mathcal{X}}}(x, x', y) = \sum_{y'} \mu_k(x, y, x', y') - \mathbf{1}_{\{X_k, X_k' = x, x'\}} \lambda_{\mathcal{X},k}(y|x)$$

and

$$\widetilde{g}_{k,\alpha_{\mathcal{Y}}}(x, y, y') = \sum_{x'} \mu_k(x, y, x', y') - \mathbf{1}_{\{Y_k, Y_k' = y, y'\}} \lambda_{\mathcal{Y},k}(x|y).$$

The following lemma provides an upper bound on each of the two $\alpha$-players.

**Lemma 5.** *With probability at least $1 - \delta$, the regret of the $\alpha_{\mathcal{X}}$-player is bounded as*

$$\text{regret}_K^{\max}(\alpha_{\mathcal{X}}^*) \leq \frac{18|\mathcal{X}|^2|\mathcal{Y}|}{(1-\gamma)^2\beta_{\mathcal{X}}} + 4\beta_{\mathcal{X}}K + \sqrt{\frac{72K}{(1-\gamma)^2}\log\frac{2}{\delta}}, \tag{33}$$

*and the regret of the $\alpha_{\mathcal{Y}}$-player is bounded as*

$$\text{regret}_K^{\max}(\alpha_{\mathcal{Y}}^*) \leq \frac{18|\mathcal{Y}|^2|\mathcal{X}|}{(1-\gamma)^2\beta_{\mathcal{Y}}} + 4\beta_{\mathcal{Y}}K + \sqrt{\frac{72K}{(1-\gamma)^2}\log\frac{2}{\delta}}. \tag{34}$$

*Proof.* We prove the claim for $\alpha_{\mathcal{X}}$, and the result for $\alpha_{\mathcal{Y}}$ will follow by symmetry. We start by noting that the gradient estimators and the gradients satisfy $\|\widetilde{g}_{k,\alpha_{\mathcal{X}}}\|_1 \leq 2$ and $\|g_{k,\alpha_{\mathcal{X}}} - \widetilde{g}_{k,\alpha_{\mathcal{X}}}\|_1 \leq 2$. Indeed, this can be verified easily as

$$\|\widetilde{g}_{k,\alpha_{\mathcal{X}}}\|_1 \leq \sum_{x,y,x',y'} \mu_k(x,y,x',y') + \sum_{x,x',y} \mathbf{1}_{\{X,X'=x,x'\}} \lambda_{\mathcal{X},k}(y|x) = 2,$$

because of the normalization of both $\mu_k$ and $\lambda_{\mathcal{X},k}$. Similarly, we have

$$\|g_{k,\alpha_{\mathcal{X}}} - g_{k,\alpha_{\mathcal{X}}}\|_1 \leq \sum_{x,x',y} \left( \mathbf{1}_{\{X,X'=x,x'\}} + \nu_{\mathcal{X}}(x,x') \right) \lambda_{\mathcal{X},k}(y|x) = 2.$$

Furthermore, $\|\alpha_{\mathcal{X}}^* - \alpha_{\mathcal{X},1}\|_\infty \leq \frac{6}{1-\gamma}$ trivially holds thanks to the definition of the domain of $\alpha_{\mathcal{X}}$ and the choice $\alpha_{\mathcal{X},1} = 0$. Finally, notice that $\Psi$ is 1-strongly convex with respect to $\|\cdot\|_2$, and thus Lemma 4 (with the choice $\widetilde{\eta} = \eta = \beta_{\mathcal{X}}$) immediately implies the claim after using the relations $\|\widetilde{g}_{k,\alpha_{\mathcal{X}}}\|_2 \leq \|\widetilde{g}_{k,\alpha_{\mathcal{X}}}\|_1 \leq 2$ and $\|\alpha_{\mathcal{X}}^* - \alpha_{\mathcal{X},1}\|_2^2 \leq \|\alpha_{\mathcal{X}}^* - \alpha_{\mathcal{X},1}\|_\infty^2 \leq \frac{36|\mathcal{X}|^2|\mathcal{Y}|}{(1-\gamma)^2}$. $\qquad\square$

### C.3.2 Regret of the $V$-player

Similarly to the $\alpha$-players, the $V$-player employs online gradient ascent on the feasible set $\mathcal{Z} = B_\infty(\frac{2}{1-\gamma})$, with entries of the gradients given in each round as

$$g_{k,V}(x,y) = \sum_{x',y'} \mu_k(x,y,x',y') - (1-\gamma)\nu_0(x,y) - \gamma \sum_{\widehat{x},\widehat{y}} \mu_k(\widehat{x},\widehat{y},x,y).$$

The following lemma gives a bound on its regret.

**Lemma 6.** *The regret of the $V$-player is bounded as*

$$\text{regret}_K^{\max}(V^*) \leq \frac{4|\mathcal{X}||\mathcal{Y}|}{\beta(1-\gamma)^2} + 2\beta K. \tag{35}$$

*Proof.* Since the $V$-player employs deterministic gradients, we will apply Lemma 4 with $1/\widetilde{\eta} = 0$, and bound the Euclidean norms of the comparator $V^*$ and the gradients. By the choice of the feasible set for $V^*$ and the choice $V_1 = 0$, we immediately have $\|V^* - V_1\|_2 \leq |\mathcal{X}||\mathcal{Y}| \|V^* - V_1\|_\infty^2 \leq \frac{4|\mathcal{X}||\mathcal{Y}|}{(1-\gamma)^2}$. Furhermore, evaluating the gradient of the Lagrangian with respect to $V$, we get

$$\|g_{k,V}\|_1 \leq (1-\gamma)\sum_{x,y} \nu_0(x,y) + (1+\gamma) \sum_{x,y,x',y'} \mu(x,y,x',y') = 2,$$

which in turn implies $\|g_{k,V}\|_2 \leq \|g_{k,V}\|_1 \leq 2$. Plugging these results in the bound of Lemma 4 concludes the proof. $\qquad\square$

### C.3.3 Regret of the $\mu$-player

The $\mu$-player plays OSMD with entropy regularization, and gradients with elements defined as

$$g_{k,\mu}(x,y,x',y') = c(x,y) - \alpha_{\mathcal{X},k}(x,x',y) - \alpha_{\mathcal{Y},k}(x,y,y') + \gamma V_k(x',y') - V_k(x,y).$$

The following bound gives a bound on the regret of this player.

**Lemma 7.** *The regret of the $\mu$-player is bounded as*

$$\text{regret}_K^{\min}(\mu^*) \leq \frac{\log(|\mathcal{X}|^2|\mathcal{Y}|^2)}{\eta} + \frac{200\eta K}{(1-\gamma)^2}.$$

*Proof.* The proof follows from noticing that the regularization function $\Psi$ is 1-strongly convex with respect to the norm $\|\cdot\|_1$, and that the dual norm of the gradients is bounded as $\|g_{k,\mu}\|_\infty \leq \frac{20}{1-\gamma}$. Indeed, this follows by upper-bounding each entry of the gradient as

$$|g_{k,\mu}(x,y,x',y')| \leq c(x,y) + |\alpha_{\mathcal{X},k}(x,x',y')| + |\alpha_{\mathcal{Y},k}(x,y,y')| + \gamma|V_k(x',y')| + |V_k(x,y)|$$

$$\leq 1 + \frac{12}{1-\gamma} + \frac{4(\gamma+1)}{1-\gamma} = \frac{17+3\gamma}{1-\gamma} \leq \frac{20}{1-\gamma}.$$

Finally, we recall the choice of $\mu_1$ being uniform over $\mathcal{X}\mathcal{Y}\mathcal{X}\mathcal{Y}$, and the standard result that the relative entropy between any distribution and $\mu_1$ is equal to $\log(|\mathcal{X}|^2|\mathcal{Y}|^2)$. $\qquad\square$

### C.3.4 Regret of the $\lambda$-players

The regret analysis of the $\lambda$-players is slightly nonstandard. Focusing on the $\lambda_{\mathcal{X}}$-player here, we note that the updates correspond to using OSMD on the decision space $\mathcal{Z} = \{\lambda : \forall x, y \ \lambda(y|x) \geq 0 \, ; \forall x \ \sum_y \lambda(y|x) = 1\}$ with the following choice of regularization function:

$$\Psi(\lambda) = \sum_x \sum_y \lambda(y|x) \log(\lambda(y|x)).$$

As we show in Lemma 10, this regularization function is 1-strongly convex with respect to the *2-1 group norm* defined for each $\lambda \in \mathcal{Z}$ as

$$\|\lambda\|_{2,1} = \sqrt{\sum_x \left( \sum_y |\lambda(y|x)| \right)^2}.$$

It is easy to verify that the corresponding dual norm is the $2 - \infty$ *group norm* defined as $\|g\|_{2,\infty} = \sqrt{\sum_x (\max_y |g(x,y)|)^2}$. We also recall that the updates make use of the following unbiased estimate of the gradient:

$$\widetilde{g}_{k,\lambda_{\mathcal{X}}}(x,y) = \mathbf{1}_{\{X=x\}} \alpha_{\mathcal{X},k}(X, X', y). \tag{36}$$

With these facts at hand, we prove the following bound on the regret of the $\lambda$-players.

**Lemma 8.** *With probability at least $1 - \delta$, the regret of the $\lambda_{\mathcal{X}}$ player and is bounded as*

$$\mathrm{regret}_K^{\max}(\lambda_{\mathcal{X}}^*) \leq \frac{|\mathcal{X}| \log |\mathcal{Y}|}{\eta_{\mathcal{X}}} + \frac{90 \eta_{\mathcal{X}} K}{(1-\gamma)^2} + \sqrt{\frac{288 |\mathcal{X}| K \log \left( \frac{2}{\delta} \right)}{(1-\gamma)^2}}. \tag{37}$$

*and the regret of the $\lambda_{\mathcal{Y}}$-player is bounded as*

$$\mathrm{regret}_K^{\max}(\lambda_{\mathcal{Y}}^*) \leq \frac{|\mathcal{Y}| \log |\mathcal{X}|}{\eta_{\mathcal{Y}}} + \frac{90 \eta_{\mathcal{Y}} K}{(1-\gamma)^2} + \sqrt{\frac{288 |\mathcal{Y}| K \log \left( \frac{2}{\delta} \right)}{(1-\gamma)^2}}. \tag{38}$$

*Proof.* We provide a complete proof for $\lambda_{\mathcal{X}}$, and note that the result for $\lambda_{\mathcal{Y}}$ is analogous. For this case, notice that Lemmas 4 and 10 suggest that we should first obtain upper-bounds on the magnitude of the gradients in terms of their $2, \infty$-group norms, and thus we first establish that

$$\|\widetilde{g}_{k,\lambda_{\mathcal{X}}}\|_{2,\infty} = \sqrt{\sum_x \left( \max_y |\mathbf{1}_{\{X=x\}} \alpha_{\mathcal{X}}(X, X', y)| \right)^2} \leq \sqrt{\sum_x \mathbf{1}_{\{X=x\}} \|\alpha_{\mathcal{X},k}\|_\infty^2} = \|\alpha_{\mathcal{X},k}\|_\infty.$$

Note that the latter is upper-bounded as $\|\alpha_{\mathcal{X},k}\|_\infty \leq \frac{6}{1-\gamma}$ by construction. Further observing that the true gradient norm can be bounded via the same argument as $\|\widetilde{g}_{k,\lambda_{\mathcal{X}}}\|_{2,\infty} \leq \frac{6}{1-\gamma}$, we also have

$$\|g_{k,\lambda_{\mathcal{X}}} - \widetilde{g}_{k,\lambda_{\mathcal{X}}}\|_{2,\infty} \leq \|g_{k,\lambda_{\mathcal{X}}}\|_{2,\infty} + \|\widetilde{g}_{k,\lambda_{\mathcal{X}}}\|_{2,\infty} \leq \frac{12}{1-\gamma}.$$

Finally, since $\lambda_{\mathcal{X},1}(\cdot|x)$ is chosen as the uniform distribution over $\mathcal{Y}$ for all $x$, we have $\mathcal{B}_\Psi(\lambda_{\mathcal{X}}^* \| \lambda_{\mathcal{X},1}) \leq |\mathcal{X}| \log |\mathcal{Y}|$, and the primal-norm distance satisfies $\|\lambda^* - \lambda\| \leq 2\sqrt{|\mathcal{X}|}$. Now, the claim follows from using Lemma 4 with $\widetilde{\eta} = \eta_{\mathcal{X}}$. $\qquad\square$

### C.4 Proof of Theorem 1

The proof of the theorem now follows from putting together Lemma 1 with the regret decomposition in Equation (32), and combining Lemmas 5–8. Taking a union bound over the two probabilistic claims of Lemma 5 and 5, this gives that the following bound holds with probability at least $1 - 2\delta$:

$$|\langle \overline{\mu}_K - \mu^*, c \rangle| \leq \frac{18 |\mathcal{X}|^2 |\mathcal{Y}|}{\beta_{\mathcal{X}} K (1-\gamma)^2} + 4\beta_{\mathcal{X}} + \sqrt{\frac{72}{K(1-\gamma)^2} \log \frac{2}{\delta}}$$

$$+ \frac{18 |\mathcal{X}| |\mathcal{Y}|^2}{\beta_{\mathcal{Y}} K (1-\gamma)^2} + 4\beta_{\mathcal{Y}} + \sqrt{\frac{72}{K(1-\gamma)^2} \log \frac{2}{\delta}}$$

$$+ \frac{4|\mathcal{X}||\mathcal{Y}|}{\beta K(1-\gamma)^2} + 2\beta$$

$$+ \frac{2\log(|\mathcal{X}||\mathcal{Y}|)}{\eta K} + \frac{200\eta}{2(1-\gamma)^2}$$

$$+ \frac{|\mathcal{X}|\log(|\mathcal{Y}|)}{\eta_{\mathcal{X}} K} + \frac{90\eta_{\mathcal{X}}}{(1-\gamma)^2} + \sqrt{\frac{288|\mathcal{X}|\log\frac{2}{\delta}}{K(1-\gamma)^2}}$$

$$+ \frac{|\mathcal{Y}|\log|\mathcal{X}|}{\eta_{\mathcal{Y}} K} + \frac{90\eta_{\mathcal{Y}}}{(1-\gamma)^2} + \sqrt{\frac{288|\mathcal{Y}|\log\frac{2}{\delta}}{K(1-\gamma)^2}}.$$

Setting $\beta_{\mathcal{X}} = \sqrt{\frac{9|\mathcal{X}|^2|\mathcal{Y}|}{2(1-\gamma)^2 K}}$, $\beta_{\mathcal{Y}} = \sqrt{\frac{9|\mathcal{X}||\mathcal{Y}|^2}{2(1-\gamma)^2 K}}$, $\beta = \sqrt{\frac{2|\mathcal{X}||\mathcal{Y}|}{(1-\gamma)^2 K}}$, $\eta_{\mathcal{X}} = \sqrt{\frac{(1-\gamma)^2|\mathcal{X}|\log|\mathcal{Y}|}{90 K}}$, $\eta_{\mathcal{X}} = \sqrt{\frac{(1-\gamma)^2|\mathcal{Y}|\log|\mathcal{X}|}{90 K}}$, $\eta = \sqrt{\frac{(1-\gamma)^2\log(|\mathcal{X}||\mathcal{Y}|)}{100 K}}$, the bound becomes

$$|\langle \overline{\mu}_K - \mu^*, c\rangle| \le \frac{12\sqrt{2|\mathcal{X}||\mathcal{Y}|}(\sqrt{|\mathcal{X}|} + \sqrt{|\mathcal{Y}|})}{(1-\gamma)\sqrt{K}} + \frac{4\sqrt{2|\mathcal{X}||\mathcal{Y}|}}{(1-\gamma)\sqrt{K}}$$

$$+ \frac{3\sqrt{10|\mathcal{X}|\log|\mathcal{Y}|}}{(1-\gamma)\sqrt{K}} + \frac{3\sqrt{10|\mathcal{Y}|\log|\mathcal{X}|}}{(1-\gamma)\sqrt{K}} + \frac{40\sqrt{\log(|X|^2|Y|^2)}}{(1-\gamma)\sqrt{K}}$$

$$+ \frac{12\sqrt{\log\frac{1}{\delta}}}{(1-\gamma)\sqrt{K}} + \frac{12\sqrt{2|\mathcal{X}|\log\frac{1}{\delta}}}{(1-\gamma)\sqrt{K}} + \frac{12\sqrt{2|\mathcal{Y}|\log\frac{1}{\delta}}}{(1-\gamma)\sqrt{K}}$$

$$= \mathcal{O}\left(\sqrt{\frac{(|\mathcal{X}||\mathcal{Y}|(|\mathcal{X}| + |\mathcal{Y}|) + |\mathcal{X}|\log\frac{|\mathcal{Y}|}{\delta} + |\mathcal{Y}|\log\frac{|\mathcal{X}|}{\delta}}{(1-\gamma)^2 K}}\right).$$

This concludes the proof.

# D   Online learning: The proof of Lemma 4

This section is dedicated to proving the general regret bound we use throughout the analysis for upper-bounding the regret of each player, Lemma 4. As mentioned in Appendix C.3, the main challenge that we need to deal with is that the comparators for some of the regret terms are data dependent, which requires some additional steps that are typically not necessary in regret analyses. For concreteness, we adapt the notation of Lemma 4 and write the regret against comparator $z^*$ as

$$\text{regret}_K(z^*) = \sum_{k=1}^K \langle g_k, z_k - z^*\rangle = \underbrace{\sum_{k=1}^K \langle \widetilde{g}_k, z_k - z^*\rangle}_{R_K} + \underbrace{\sum_{k=1}^K \langle g_k - \widetilde{g}_k, z_k - z^*\rangle}_{M_K},$$

where in the second equality we also added some terms corresponding to the stochastic gradient $\widetilde{g}_k$. Here, the first term $R_K$ corresponds to the regret of the online learning algorithm on the sequence of stochastic gradients $\widetilde{g}_k$, which can be upper-bounded using standard tools of online learning. For the second term, notice that the stochastic gradient satisfies $\mathbb{E}[\widetilde{g}_k | \mathcal{F}_{k-1}] = g_k$, and thus if $z^*$ is independent of the sequence of stochastic gradients, the second term $M_K$ in the above decomposition is a martingale. However, this is no longer true if $z^*$ is statistically dependent on the sequence. In order to account for this, we adopt an elegant technique by Rakhlin and Sridharan [2017] to control the resulting sequence of dependent random variables[1]. In particular, we introduce a second online learning algorithm for the sake of analysis, and use its regret bound to account for the additional error terms in the above decomposition. For sake of concreteness, we define the sequence of decisions made by this algorithm by setting $\widetilde{z}_1 = z_1$ and updating the parameters recursively via a mirror

---

[1]This technique is commonly attributed to Nemirovski et al. [2009], but we find the connection with Rakhlin and Sridharan [2017] more illuminating. Otherwise, we learned this proof technique from Neu and Okolo [2024].

descent scheme analogous to the one underlying the sequence $z_k$:

$$\widetilde{z}_{k+1} = \arg\min_{z \in \mathcal{Z}} \left\{ \langle g_k - \widetilde{g}_k, z \rangle + \frac{1}{\eta} \mathcal{B}_\Psi(z \| z_k) \right\}.$$

Using this notation, the regret of the original algorithm can be rewritten as follows:

$$\sum_{k=1}^{K} \langle g_k, z_k - z^* \rangle = \underbrace{\sum_{k=1}^{K} \langle \widetilde{g}_k, z_k - z^* \rangle}_{R_K} + \underbrace{\sum_{k=1}^{K} \langle g_k - \widetilde{g}_k, z_k - \widetilde{z}_k \rangle}_{\widetilde{M}_K} + \underbrace{\sum_{k=1}^{K} \langle g_k - \widetilde{g}_k, \widetilde{z}_k - z^* \rangle}_{\widetilde{R}_K}.$$

Thanks to this construction, the term $\widetilde{M}_K$ is a martingale and $\widetilde{R}_K$ is the regret of the auxiliarly online learning algorithm in the newly defined online learning game.

For the concrete proof of Lemma 4, we will make use of the following classic result regarding the regret of mirror descent.

**Lemma 9.** *Let $z \in \mathcal{Z}$ and assume that $\Psi$ is $\lambda$-strongly convex with respect to some norm $\|\cdot\|$ whose dual is denoted by $\|\cdot\|_*$. Consider the sequence with an arbitrary $u_1 \in \mathcal{Z}$ and all subsequent iterates defined as*

$$u_{k+1} = \arg\min_{z \in \mathcal{Z}} \left\{ \langle v_k, z \rangle + \frac{1}{\omega} \mathcal{B}_\Psi(z \| u_k) \right\},$$

*where $a_k$ is an arbitrary sequence in $\mathcal{Z}^*$ and $\omega > 0$. Then, for any $u^* \in \mathcal{Z}$, the sequence $(u_k)_k$ produced by OSMD satisfies the following bound:*

$$\sum_{k=1}^{K} \langle a_k, u_k - u \rangle \leq \frac{\mathcal{B}_\Psi(u^* \| u_1)}{\omega} + \frac{\omega}{2\lambda} \sum_{k=1}^{K} \|a_k\|_*^2. \tag{39}$$

The proof is standard and can be found in many textbooks—for concreteness, we refer to Theorem 6.10 of Orabona [2019]. To proceed, we apply this lemma to the standard sequence of iterates in our setting with $a_k = \widetilde{g}_k$ and $\omega = \eta$ to bound $R_K$ and once again with $a_k = g_k - \widetilde{g}_k$ and $\omega = \widetilde{\eta}$ to bound $\widetilde{R}_K$. Finally, we use the Hoeffding–Azuma inequality (Lemma 11) to control the remaining term as

$$\widetilde{M}_K = \sum_{k=1}^{K} \langle g_k - \widetilde{g}_k, z_k - \widetilde{z}_k \rangle \leq C \sqrt{2 \sum_{k=1}^{K} \|g_k - \widetilde{g}_k\|^2 \log \frac{1}{\delta}}$$

with probability at least $1 - \delta$. Indeed, notice that under the condition $\max_{z,z' \in \mathcal{Z}} \|z - z'\| \leq C$ each term satisfies $|\langle g_k - \widetilde{g}_k, z_k - \widetilde{z}_k \rangle| \leq C \|g_k - \widetilde{g}_k\|_*$, which allows using Lemma 11 with $c_k = 2C \|g_k - \widetilde{g}_k\|_*$. Putting these results together concludes the proof of Lemma 4.

# E Technical Lemmas

**Lemma 10.** *The function $\Psi(p) = \sum_{j=1}^{J} \sum_{i=1}^{I} p(i|j) \log p(i|j)$ is 1-strongly convex with respect to the 2-1 group norm $\|p\|_{2,1} = \sqrt{\sum_{j=1}^{J} \left( \sum_{i=1}^{I} |p_{i|j}| \right)^2}$ on the set $\mathcal{Z} = \{ p \in \mathbb{R}^{I \times J} : p(i|k) \geq 0 (\forall i, j), \sum_{i=1}^{I} p(i|j) = 1 (\forall j) \}$.*

*Proof.* We first note that, by standard calculations, the Bregman divergence induced by $\Psi$ is

$$B_\Psi(p \| q) = \sum_{j=1}^{J} \sum_{i=1}^{I} p(i|j) \log \frac{p(i|j)}{q(i|j)}.$$

Now, by Pinsker's inequality, we have that

$$B_\Psi(p \| q) \geq \frac{1}{2} \sum_{j=1}^{J} \|p(\cdot|j) - q(\cdot|j)\|_1,$$

which is equivalent to the statement of the lemma. $\square$

**Lemma 11.** *(Hoeffding–Azuma inequality, see, e.g., Lemma A.7 in Cesa-Bianchi and Lugosi 2006)*
*Let $(Z_k)_k$ be a martingale with respect to a filtration $(\mathcal{F}_k)_k$. Assume that there are predictable processes $(A_k)_k$ and $(B_k)_k$ and positive constant $(c_k)_k$ such that for all $k \geq 1$, almost surely,*

$$A_k \leq Z_k - Z_{k-1} \leq B_k \quad and \quad B_k - A_k \leq c_t.$$

*Then, for all $\epsilon > 0$,*

$$\mathbb{P}\left[Z_t - Z_0 \geq \epsilon\right] \leq \exp\left(-\frac{2\epsilon^2}{\sum_{i=1}^t c_i^2}\right), \tag{40}$$

*or equivalently for all $\delta \in (0,1)$*

$$\mathbb{P}\left[Z_t - Z_0 \geq \sqrt{\frac{\left(\sum_{i=1}^t c_i^2\right)\log(\frac{1}{\delta})}{2}}\right] \leq \delta. \tag{41}$$

**Lemma 12.** *The occupancy measure $\nu_{\mathcal{X}} \in \mathbb{R}_+^{\mathcal{X} \times \mathcal{X}}$ of the Markov chain $M_{\mathcal{X}}$ is uniquely defined by the two sets of equations*

$$\sum_{x'} \nu_{\mathcal{X}}(x, x') = \gamma \sum_{x''} \nu_{\mathcal{X}}(x'', x) + (1 - \gamma)\nu_{0,\mathcal{X}}(x) \quad (\forall x), \tag{42}$$

$$\nu_{\mathcal{X}}(x, x') = P_{\mathcal{X}}(x'|x) \sum_{x''} \nu_{\mathcal{X}}(x, x'') \quad (\forall x, x'). \tag{43}$$

*Proof.* Using the definition of the occupancy measure $\nu_{\mathcal{X}}$ we obtain

$$\nu_{\mathcal{X}}(x, x') = (1 - \gamma) \sum_{t=0}^{\infty} \gamma^t \mathbb{P}\left[X_t = x, X_{t+1} = x'\right]$$

$$= (1 - \gamma) \sum_{t=0}^{\infty} \gamma^t P_{\mathcal{X}}(x'|x)\mathbb{P}\left[X_t = x\right]$$

$$= P_{\mathcal{X}}(x'|x)\left((1 - \gamma)\nu_{0,\mathcal{X}}(x) + (1 - \gamma)\sum_{t=1}^{\infty}\gamma^t\mathbb{P}\left[X_t = x\right]\right)$$

$$= P_{\mathcal{X}}(x'|x)\left((1 - \gamma)\nu_{0,\mathcal{X}}(x) + \gamma\sum_{x''}(1 - \gamma)\sum_{t=1}^{\infty}\gamma^{t-1}\mathbb{P}\left[X_{t-1} = x'', X_t = x\right]\right)$$

$$= P_{\mathcal{X}}(x'|x)\left((1 - \gamma)\nu_{0,\mathcal{X}}(x) + \gamma\sum_{x''}\nu_{\mathcal{X}}(x'', x)\right),$$

where we used the stationarity of the transition kernel $P_{\mathcal{X}}$, the definition of $\nu_{0,\mathcal{X}}$, the law of total probability, and the stationarity of the Markov chain to recognize $\nu_{\mathcal{X}}(x'', x)$ in the last step. Summing the previous equation over $x'$ yields (42), and substituting (42) into the previous equation yields (43).

In order to show that the solution $\nu_{\mathcal{X}}$ to (42) and (43) is unique, we introduce the notation $\xi_{\mathcal{X}}$ as $\xi_{\mathcal{X}}(x) = \sum_{x'} \nu_{\mathcal{X}}(x, x')$ for each $x$. Substituting (43) into (42) yields

$$\xi_{\mathcal{X}}(x) = \gamma \sum_{x''} P_{\mathcal{X}}(x|x'')\xi_{\mathcal{X}}(x'') + (1 - \gamma)\nu_{0,\mathcal{X}}(x) \quad (\forall x).$$

By defining $\xi_{\mathcal{X}}$ and $\nu_{0,\mathcal{X}}$ as vectors and $P_{\mathcal{X}}$ as a matrix, we can write this system of equations in matrix form as $\xi_{\mathcal{X}} = \gamma P_{\mathcal{X}}\xi_{\mathcal{X}}^{\mathsf{T}} + (1 - \gamma)\nu_{0,\mathcal{X}}$, or equivalently, $(I - \gamma P_{\mathcal{X}}^{\mathsf{T}})\xi_{\mathcal{X}} = (1 - \gamma)\nu_{0,\mathcal{X}}$. Since $P_{\mathcal{X}}$ is a positive matrix with spectral radius 1, the Perron–Frobenius theorem applies and the matrix $(I - \gamma P_{\mathcal{X}})$ is invertible. Hence there exists a unique solution $\xi_{\mathcal{X}} = (1 - \gamma)(I - \gamma P_{\mathcal{X}}^{\mathsf{T}})^{-1}\nu_{0,\mathcal{X}}$, which together with (43) implies that $\nu_{\mathcal{X}}$ is uniquely defined as $\nu_{\mathcal{X}}(x, x') = P_{\mathcal{X}}(x'|x)\xi_{\mathcal{X}}(x)$. $\square$

# F Additional details on experiments

In this appendix we present further details about the experiments included in the main text, along with some additional empirical results. Along the way, we will also provide some further comments on best practices when implementing **SOMCOT**, including recommended hyperparameter settings.

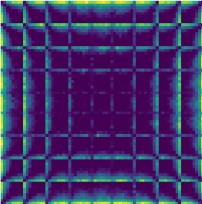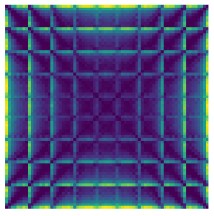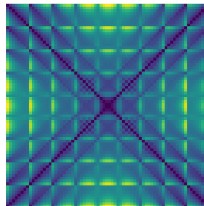

Figure 3: Distance matrices between instances after running **SOMCOT** for 1000 and 10000 steps, and the ground truth obtained via Sinkhorn Value Iteration.

## F.1 Similarity metrics between parametric Markov chains

This experiment serves to illustrate the ability of bisimulation metrics to capture intuitive similarities between stochastic processes, as well as the empirical behavior of **SOMCOT** when used to approximate such similarity metrics based on data. To this end, we generated several random walk instances from the same family as used in the experiments in the main body (described in detail in Appendix F.3). For this experiment, we let $\mathcal{X} = \mathcal{Y} = \{1, 2, \ldots, n\}$ with $n = 1$ and set the block size as $B = 1$. Deviating from the setup described in Appendix F.3, we set the reward function as $r(1) = r(n) = 1$ for both extremes of the state space, which induces a symmetry on the state space. For generating the set of environments, we varied the initial states $x_0$ and $y_0$ between $\{2, 3, \ldots, n-1\}$ and the bias parameter $\theta$ in the set $\{0.05, 0.1, 0.15, \ldots, 0.95\}$, thus resulting in 72 different instances. We then computed pairwise distances between these instances using **SOMCOT** with various sample sizes, and compared the results with the ground truth (computed by the Sinkhorn Value Iteration method of Calo et al. 2024). Figure 3 shows the similarity matrices obtained by these methods, showing that the distances computed by **SOMCOT** successfully capture the structure of the problem: even though the exact numerical values of the true distances are not approximated very accurately, the qualitative picture obtained by **SOMCOT** is very similar to the ground truth. In particular, the symmetry induced our choice of reward function is clearly visible with the matrix being symmetric along the counter-diagonal as well as the main diagonal.

## F.2 Practical implementation details

Being a primal-dual method, **SOMCOT** is not as easy to tune as a common stochastic optimization algorithm. There are several implementation details that one needs to design carefully in order to make sure that the algorithm behaves in a stable way and outputs good estimates. This section describes our experience working with **SOMCOT**, and provides practical guidance for implementation.

**SOMCOT** has one tunable parameter per optimization variable: a positive learning rate that controls the magnitude of the updates during optimization. While our theoretical analysis suggests some specific values for these learning rates to guarantee convergence, such values are typically too conservative (as is common in stochastic optimization). In practice, using larger learning rates can significantly reduce the number of iterations needed to reach good solutions. Since our problem involves optimizing six variables, this leads to six separate hyperparameters, which makes tuning a grueling task. To address this, we tie some of the learning rates together: all primal variables share a single learning rate denoted by $\eta$, and all dual variables share another one denoted by $\beta$.

Moreover, we observed that in practice using a fixed value for the dual learning rate $\eta$ often made it difficult to achieve stable convergence across different problem instances. To address this, we introduced a decaying learning-rate scheme of the form $\eta_k = \frac{\eta_0}{\sqrt{1+ak}}$, where $k$ is the index of the current iteration and $a > 0$ is a tunable parameter. This decay helps balance the need for large updates in early iterations with the stability required for convergence in later stages.

Figure 4 illustrates the performance of the algorithm under different learning rate settings. This shows that, even given the above choices, it is not easy to pick hyperparameters that work uniformly well across problem instances. Even for a single instance, the combination of $\eta_0$ and $\beta$ that leads to the best performance requires careful hyperparameter search. In order to understand the behavior of **SOMCOT** under different parameter choice, it is helpful to remember the roles of the primal and dual variables,

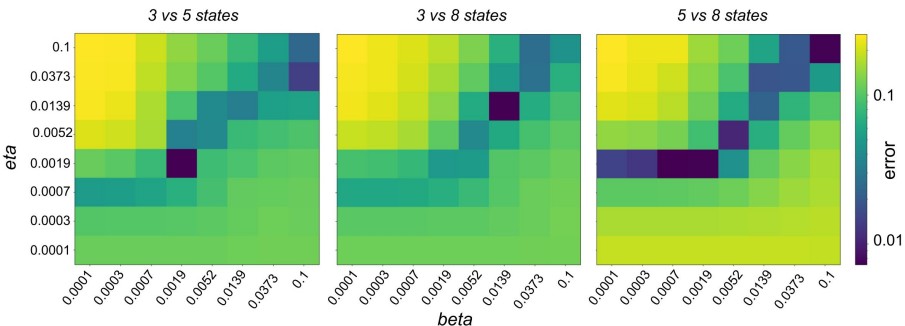

Figure 4: The influence of the ratio between $\eta$ and $\beta$ on the convergence of **SOMCOT** for different chain sizes. Error and learning rates are shown on a logarithmic scale. To produce this plot, a decay rate of $a = 0.001$ was used for $\eta$. No decay was applied on $\beta$.

and in particular that the dual variables serve to penalize the primal variables for violating the primal constraints. Thus, a value of $\eta$ that is too high relative to $\beta$ leads to large constraint violations, resulting in $\mu$ values that yield very small distances but fall outside the feasible set. Ultimately, setting $\beta$ too small results in gross underestimation of the true distance. Thus, whenever one sees distance estimates that are suspiciously close to zero, the value of $\beta$ should be increased or the value of $\eta$ be decreased. The opposite scenario produces the inverse effect: a $\beta$ that is too large relative to $\eta$ causes the dual variables to update too quickly, leading to a resulting distance that overestimates the actual value. This issue can largely be mitigated by decaying $\eta$ while keeping $\beta$ constant (as described above). In our experience, it is often better to pick a large initial value for $\eta$: while this typically leads to a rapid drop of the distance estimate to zero, the estimates eventually start increasing and converge toward the true cost.

Theorem 1 provides guarantees for the averaged output $\overline{\mu}_K = \frac{1}{K} \sum_{k=1}^{K} \mu_k$, where $\mu_k$ is the value of $\mu$ obtained at iteration $k$. This is commonly required for algorithms based on regret analysis, at least for the theoretical guarantees to go through. In typical applications of stochastic optimization, this averaging step is not strictly necessary and the final iteration can perform well enough. However, this is typically not the case for primal-dual algorithms like **SOMCOT**, where iterate averaging often makes a big difference to the stability of algorithms. This is true in our case too: without averaging, the iterates typically fluctuate quite wildly around the optimum. Averaging makes the estimates much more stable, and is thus strongly recommended (even if only for the last half of the iterates or less).

Finally, we note that all our experiments have made use of i.i.d. transitions sampled from the occupancy measures of the two chains. This falls in line perfectly with the theory, but may be impractical in applications where transitions may be dependent or be sampled from undiscounted trajectory distributions. While we have not experimented with such data, we believe that **SOMCOT** should be able to deal with it as long as efforts are made to break the correlations between the consecutive samples, for instance by sampling the transitions randomly from a buffer (instead of processing them in their original order). In our experiments, we have sometimes made use of minibatch updates, which can affect computational efficiency and stability, but no major impact on the overall convergence properties has been observed. We display all hyperparameter choices we have made in the experiments in Table 1.

### F.3 Details about the environments

Our experiments made use of two families of Markov chains: a collection of parametrized random walks, and several instances of the classic "inverted pendulum" environment. We describe the details of these settings below.

**Parametrized random walks.** We consider a one-dimensional random walk over a finite state space $\mathcal{X} = \{1, 2, \ldots, n\}$ with biased transitions. A transition from state $x$ moves to $x + 1$ with probability $\theta \in [0, 1]$ and $x - 1$ with probability $1 - \theta$. States $1$ and $n$ are "sticky walls": the process remains there with probability $0.9$ or moves to the neighboring state with probability $0.1$.

| Experiment | $\eta_0$ | $a$ | $\beta$ | $b$ | $\gamma$ |
|---|---|---|---|---|---|
| Figure 1 | 40 | 0 | 0.2 | 1 | 0.99 |
| Figure 3 | 20 | 0 | 0.5 | 1 | 0.99 |
| Figure 2a | 0.1 | 0.001 | 0.5 | 8 | 0.95 |
| Figure 2b | 0.1 | 0.05 | 0.2 | 16 | 0.95 |

Table 1: Table summarizing our hyperparameter choices for each experiment. Recall that the learning rates follow the decaying scheme $\eta_k = \frac{\eta_0}{\sqrt{1+ak}}$, and the minibatch size is denoted by $b$.

Additionally, we define a reward function on the state space, with values $r(1) = 1$, $r(n) = -1$, and $r(x) = 0$ for all $x \in 2, \ldots, n-1$. The initial state distribution is a Dirac measure on $x = 1$. To produce the plot shown in Figure 2a, we generate a low-dimensional chain $M_{\mathcal{X}}$ with bias $\theta = 0.5$ following this setting. Then, we produce a set of chains $M_{\mathcal{Y}} \in \mathcal{B}$, each of them with a different bias parameter. In addition, all $M_{\mathcal{Y}}$ are augmented with an additional irrelevant noise variable, producing $B$ observations per each latent state in $\mathcal{X}$. Formally, $M_{\mathcal{Y}}$ is a Markov chain on the state space $\mathcal{Y}$ equal to $\mathcal{X} \times \{1, 2, \ldots, B\}$. The cost between $x \in \mathcal{X}$ and $y \in \mathcal{Y}$ is given by $c(x, y) = |r(x) - r(y)|$, reflecting the absolute difference in rewards between the states.

**Inverted pendulum.** We begin by training a near-optimal policy using DDPG, a widely known Deep RL algorithm, in the standard *Pendulum-v1* environment, which is then used to induce a Markov chain. One could use any policy, but using a near-optimal policy produces richer dynamics (e.g., using a random policy in the *Pendulum-v1* environment reduces the effective state space to the surroundings of the initial state). Once a policy is fixed, we discretized each state variable of the environment into $n$ bins. Since the *Pendulum-v1* environment has 2 variables (angle $\theta$ and angular velocity $w$), the resulting state space has $n^2$ states. In our experiments, we have chosen $n = 7$, which resulted in a total of 49 states. Note that due to the discretization, the resulting stochastic process is no longer a Markov chain, as the states are no longer sufficient to predict the distribution of the next state. Nevertheless, the conducted experiments follow the same principle as the aforementioned random walks: We will compare the (approximate) Markov chain $M_{\mathcal{X}}$ of discretized observations with a set of parametrized models $M_{\mathcal{Y}}$. The parameter governing the dynamics the acceleration constant $g$, capturing the effect of gravity. We set the default value $g = 9.8$ in $M_{\mathcal{X}}$, and choose values in $[0.4, 20]$ in the model set. The cost function is given by $c(x, y) = |r(x) - r(y)|$, where $r(x)$ is the average reward in bin $x$ (computed from all samples that fell into bin $x$ along a long simulated trajectory).

