# OpenReview forum: "Distances for Markov chains from sample streams"
_NeurIPS.cc/2025/Conference — NeurIPS 2025 poster_

### Official Review · Reviewer_KfLQ · 2025-06-23

**Clarity:** 3
**Significance:** 3
**Originality:** 3
**Rating:** 4
**Confidence:** 3

**Summary:**

This paper proposes a new stochastic optimization method for computing bisimulation distances between finite Markov chains using only sample trajectories, without requiring knowledge of the transition kernels. They extend results from   Calo et al. [2024], where computation of this metric required explicit transition kernels. The method is grounded in a novel linear programming (LP) formulation that characterizes the so-called bicausal couplings, enabling an estimation of the optimal transport distance between Markov processes. The authors introduce a saddle-point stochastic optimization algorithm (SOMCOT), provide rigorous sample complexity guarantees, and validate the approach with experiments. The paper builds upon recent work connecting bisimulation metrics with optimal transport distances and aims to generalize this theory to settings with limited observability.

**Questions:**

* Q1:  page 3, line 3: Why do the authors refer to the processes as “stationary,” despite having deterministic initial states? Was it meant “ergodic” or “time-homogeneous”?

* Q2: in “Preliminaries” there are notations $\mathcal{X}^\infty$ and $\mathcal{X}^\mathbb{N}$? Are they the same?


* Q3: How does the proposed bicausal coupling compare to classical Markovian couplings? Could this be clarified more explicitly (is one a subset of the other?) (for Markovian vs non-Markovian couplings see e.g. [R1])


* Q4: Could the authors provide a concrete example (e.g., from RL or model selection) where computing this distance would lead to a practically useful decision?


* Q5: Why are the experiments focused solely on synthetic examples? Have any real-world applications been attempted or considered?


* Q6:  The proposed formulation removes the need to sample from transition kernels \( P_X(\cdot|x) \), \( P_Y(\cdot|y) \), and instead relies on samples from the joint distributions over consecutive states. Could the authors clarify what assumptions are made about the availability of such samples? For example, if only a single long trajectory per process is available, under what mixing or subsampling conditions can these empirical joint distributions be used reliably in the stochastic optimization algorithm?

* Q7: Each SOMCOT iteration costs $\Theta(|X|^2|Y|^2)$ time and $O(|X|^2|Y|^2)$  memory, while the sample-complexity bound scales only with $|X||Y|(|X|+|Y|)$.  For state spaces beyond a few hundred states this becomes prohibitive. Do the authors foresee low-rank, sparse, or neural parameterizations that could make the algorithm practical? A brief discussion would strengthen the experimental narrative.

* Q8: The distance is defined with a discount factor $\gamma\in(0,1)$. How sensitive is the estimator to the choice of $\gamma$in practice? Are there guidelines (or theory) for selecting $\gamma$ when the horizon of interest is finite or unknown?

[R1] Czumaj, A., Vöcking, B. (2014). Thorp Shuffling, Butterflies, and Non-Markovian Couplings. In: Esparza, J., Fraigniaud, P., Husfeldt, T., Koutsoupias, E. (eds) Automata, Languages, and Programming. ICALP 2014. Lecture Notes in Computer Science, vol 8572. Springer, Berlin, Heidelberg.

**Ethical Concerns:**

["NO or VERY MINOR ethics concerns only"]

**Final Justification:**

I maintain my borderline accept rating, but acknowledge that other reviewers - - likely more familiar with the application domain -- see stronger potential in this work. The rebuttal’s clarification of the RL motivation, the role of bisimulation metrics in representation learning, and the relationship between bicausal and Markovian couplings strengthen the case, and I believe the paper should be accepted.

**Limitations:**

yes

**Quality:**

4

**Strengths And Weaknesses:**

**Strengths:**
* **Theoretical rigor**: The proposed method is based on a solid and novel LP formulation of bisimulation metrics, with full theoretical guarantees and a clean stochastic primal-dual algorithm.

* **Nontrivial contribution**: Reformulating the problem to eliminate explicit dependence on transition kernels and make it sample-based is meaningful and potentially useful.

* **Relation to OT and RL**: The work connects two active lines of research—optimal transport and bisimulation in RL—and contributes a new theoretical perspective.


**Weaknesses:**

* **Unclear practical motivation**: Although the paper highlights real-world relevance (e.g., only having access to sample trajectories), it lacks concrete scenarios where this distance is actually needed. No concrete application or decision-making scenario is provided where this distance is clearly useful.

* **Limited experimental insight**: The experimental section includes illustrative examples (e.g., block Markov chains, parameterized random walks), but it is unclear what questions these experiments are intended to answer or what insight is gained. No direct comparison with competing estimators or baselines is provided.

* **No clear real-world application**: It remains open how this framework would be applied in, for instance, reinforcement learning, model comparison, or structure discovery—despite several such references in the introduction.


*Comments*
* While the paper focuses on optimal transport-based distances between Markov processes, it might benefit from briefly situating itself within the broader literature on estimating or testing properties of Markov chains from a single trajectory. You should e.g. consider mentioning:
    * [a] Hsu, D., Kontorovich, A., and Szepesvári, C. (2019). Mixing Time Estimation in Reversible Markov Chains from a Single Sample Path. The Annals of Applied Probability, 29(4), 2439–2480.
    * [b] Daskalakis, C., Dikkala, N., Kamath, G., and Tixier, R. (2022). Identity Testing of Reversible Markov Chains. AISTATS 2022,  151, 1516–1540.
    * [c] Wolfer, G., and Kontorovich, A. (2019). Estimating the Mixing Time of Ergodic Markov Chains. Proceedings of the 32nd Annual Conference on Learning Theory (COLT), PMLR 99, 1–40.
    * [d]  Wolfer, G. (2020). Mixing Time Estimation in Ergodic Markov Chains from a Single Trajectory with Contraction Methods. Proceedings of the 31st International Conference on Algorithmic Learning Theory (ALT), PMLR 117, 1–15.

* **To clarify**: my main concern relates to the lack of a clearly articulated practical use case for the proposed method. However, since optimal transport and RL  are not my  area of expertise, it's possible that I’m overlooking applications that others may appreciate. If this point is addressed or judged less important by other reviewers, I would be glad to see the paper accepted.

---

> ### Author Rebuttal · Authors · 2025-07-29
>
> Thank you for your critical reading of our work and your thoughtful comments!
>
> Regarding your remarks listed under “weaknesses”: we concede that we may have not spent too much effort on explaining the practical relevance of learning bisimulation metrics (a few references aside). This might have been because bisimulation metrics are very popular for the purpose of representation learning in our “home community” of RL, and their usefulness is widely acknowledged in this area—see, e.g., the well-known work of Zhang et al. (2021) cited in our submission. That work clearly demonstrates the usefulness of bisimulation metrics, and in fact has motivated some of us to explore this topic in the first place. Since then, our main focus has been developing theoretically grounded methods for estimating bisimulation metrics, complementing the existing range of works that almost exclusively use heuristics for this purpose. We will highlight this motivation more clearly in the final version of the paper.
>
> As for the experiments: our purpose was to demonstrate some original ways in which bisimulation metrics could be used for representation learning. We (and other readers) have found these to be interesting. For instance, we would not be able to think of another tractable way of learning an “encoder-decoder” pair illustrated on Figure 1, and we believe that this could be turned into a useful method for representation learning. We do agree though that there is no concrete hypothesis that is being tested in these experiments, and we don’t make claims about beating any other baselines (mostly because there are no existing baselines to compare with). We simply believed that the theoretical claims made in the paper are strong enough to grant us a bit of a creative license with the experiments. We hope that you don’t disagree completely with our stylistic choices.
> In the final version, we will also make sure to mention other works on testing properties of Markov chains — this literature is indeed relevant! We would concretely highlight the potential of our method for estimating mixing times of Markov chains by exploiting the link between Markovian couplings and mixing times.
>
> Finally, thank you for your detailed comments! We will update the paper to address all of them; we respond briefly to each point below.
>
> - **Q1:** Indeed, this should have been “time-homogenous”.
> - **Q2:** Thanks for catching this, these two notations indeed refer to the same idea.
> - **Q3:** Markovian couplings form a subset of bicausal couplings, see, e.g. the work of Moulos (2021) on the same problem as we consider. Thank you for the pointer to the “Thorp shuffling, etc” paper, it looks very interesting!
> - **Q4:** See our response above.
> - **Q5:** For now, our method is limited to small finite state spaces; we hope to extend it towards larger problems in future work. As noted above, we hoped that the experiments would pique the interest of the reader despite being limited to relatively small problems.
> - **Q6:** As explained in Appendix B.4, obtaining samples from the occupancy measures requires the ability to reset to the initial state. If such resets are not available, optimizing the discounted objective we consider is probably not a good idea in the first place, and it might be better to consider the undiscounted case (roughly corresponding to sending $\gamma$ to 1). We believe that our analysis can be adapted to this case under commonly considered mixing conditions on the process, but we opted to focus on the discounted case for simplicity here.
> - **Q7:** This is one of the most interesting questions we aim to address in future work. We note that we find it unreasonable to expect that the sample and computational complexity can be improved without further assumptions on the processes. Identifying the correct assumptions that enable such improvements is a very important question, and we indeed have our eyes on structures such as low-rank or sparse transition matrices.
> - **Q8:** The discount factor $\gamma$ is part of the problem definition, and its “correct” choice always depends on the specifics of the application one has in mind. The same considerations apply as in the broader field of dynamic programming and reinforcement learning: small discount factors make the objective easier to optimize but the solution to be more myopic, whereas large discounts can capture the dynamic more meaningfully at the expense of making the optimization process more challenging. The factor $1/(1-\gamma)$ can be thought of as the time scale at which the differences between the two chains are effectively taken into account (the “effective horizon”). For time scales that are not known a priori, we believe that it may be more meaningful to consider the undiscounted case mentioned above.

---

> > ### Comment · Reviewer_KfLQ · 2025-08-03
> > **About responses**
> >
> > Thank you for addressing my questions. I remain positive about the paper overall.

---

> > > ### Author Response · Authors · 2025-08-05
> > >
> > > Thank you for your response & your appreciation of our work!
> > > Best regards,
> > > Authors

---

### Official Review · Reviewer_bjRP · 2025-06-26

**Clarity:** 4
**Significance:** 4
**Originality:** 4
**Rating:** 6
**Confidence:** 3

**Summary:**

The goal of this work is to approximate the Wasserstein distance between two distributions over an infinite-dimensional data space, from samples. Specifically, these distributions are discrete-time Markov processes.

The idea is as follows:
- Classicaly write the Wasserstein divergence as minimizing a cost over couplings between the two distributions (Eq 1)
- Decompose the distributions autoregressively using the Markov transition probabilities or occupancy measures (Eq 2)
- Characterize the occupancy measures by three constraints (Eqs 6-7-8)
- Dualize these constraints (Eq 9)
- Solve the resulting min-max problem using a stochastic primal-dual method (Algorithm 1).

The authors provide computational and sample complexity guarantees (Section 4) and experimental validation (Section 5).

**Questions:**

Q1. Is this analysis amenable to continuous-time Markov chains, so as to compute the Wasserstein distance between two path measures?

Q2. This work shows how to *estimate* the Wasserstein divergence between two Markov chains from samples, using a primal-dual optimization algorithm to solve a min-max formulation of the divergence. Suppose we additionally wanted *optimize* through the divergence, for example if one Markov chain is the "ground truth" and the other Markov chain is a "parametric model". Do the authors think  that the extra minimization would be numerically stable?

**Ethical Concerns:**

["NO or VERY MINOR ethics concerns only"]

**Final Justification:**

I would like to maintain my score of "strong accept".

I believe the authors' contribution is interesting, important and well-explained. I have read through their answers to other reviewers and am satisfied that they are addressing in good faith the issues that were brought up.

In particular, I saw some concerns on computational complexity and more extensive experiments. I do not believe they are central to this work, whose main contribution is proposing a relatively novel theoretical framework for computing optimal transport distances between Markov Chains and illustrating that it is applicable on reasonable experiments. How to further scale this algorithm would be the topic of another paper in my opinion. Also, in my understanding, there are not many competing algorithms to benchmark against nor is the point to claim a SOTA result. Rather, the experiments serve as a proof-of-concept that this quite theoretical framework can be used in practice.

**Limitations:**

Yes.

**Paper Formatting Concerns:**

None.

**Quality:**

4

**Strengths And Weaknesses:**

**Strengths**. The writing is exceptionally clear. The main steps are clearly conveyed (see summary). Furthermore, the writing and notations make the parallels clear between this setup and the finite-dimensional case. It is appreciable to have a a theoretical analysis of the computational and sample complexity and Figure 2.b. is a particularly convincing use-case.

**Weaknesses**. None in particular.

---

> ### Author Rebuttal · Authors · 2025-07-29
>
> Thank you for your positive evaluation of our work and thoughtful comments!
>
> Regarding your questions:
>
> 1. In principle, many of the objects in our formulation (and particularly occupancy couplings) do generalize to continuous time and state spaces, but we believe that adapting our algorithms to such settings would be extremely challenging (at least for us, who are mostly computer scientists). For instance, the constraints in the LP would need to be replaced by differential equations, and we dread to think about the conditions required for Lagrangian duality to hold in this scenario. We leave the investigation of this setting to more qualified colleagues.
> 2. Optimizing our distance estimates with respect to parametric models is one of the most exciting questions we would like to explore in the future! Note that there is a clear path towards doing this, at least conceptually: the dual variables at the optimum give the derivatives with respect to the occupancy measures, which themselves are differentiable with respect to the kernels. We have not considered proving guarantees about the accuracy of our estimates of the dual variables so far (which would give an idea about the accuracy of a gradient-based optimization method for optimizing the representations), but this is definitely on our long list of topics to explore! Thank you for the suggestion!

---

> ### Comment · Reviewer_bjRP · 2025-08-03
> **Answer to authors**
>
> Thank you for your answer which I agree with. I too remain positive about the paper overall.

---

> > ### Author Response · Authors · 2025-08-05
> >
> > Thank you for your response & your appreciation of our work!
> > Best regards,
> > Authors

---

### Official Review · Reviewer_52pg · 2025-07-03

**Clarity:** 3
**Significance:** 3
**Originality:** 3
**Rating:** 4
**Confidence:** 4

**Summary:**

This paper studies the problem of estimating the adapted (bicausal) Wasserstein distance between two finite-state Markov chains from sample trajectories. Building on the observation that the bisimulation metric used in reinforcement learning coincides with the adapted OT distance under discounted cost, the authors propose a stochastic primal-dual algorithm (SOMCOT) that operates using only sample access to the underlying chains.

A key contribution is a new linear programming formulation over occupancy couplings that eliminates dependence on the transition kernels, making the problem amenable to stochastic approximation. The algorithm performs mirror descent on the primal variables and projected gradient ascent on the duals. A non-asymptotic error bound of order O\left( \frac{1}{\sqrt{K}(1 - \gamma)} \right) is established, with dimension dependence matching that of standard OT estimators.

Empirical results illustrate the method’s capacity to recover latent state structure and perform model selection from sample data. The formulation is theoretically sound and avoids the heuristics common in earlier deep RL applications of bisimulation. While the algorithm is framed in the language of bisimulation, the underlying quantity is precisely the discounted adapted Wasserstein distance, and the contribution is best viewed as a sample-based estimation method for this object.

**Questions:**

1.	Comparison with Calo et al. (2024):
The proposed LP formulation appears closely related to the one in Calo et al. (2024), replacing conditional transition kernels P(x{\prime}|x) with the marginal occupancy distribution \nu_X(x, x{\prime}). While the authors argue that their formulation is more suitable for the sample-access setting, conditional distributions can also be estimated from trajectories. Could the authors clarify precisely why their formulation offers an advantage over that of Calo et al., especially since this work builds directly upon it? A more careful comparison—both conceptual and empirical—seems warranted.
	2.	Sampling Cost under Discounting:
The algorithm requires samples from the distribution
(1 - \gamma) \sum_{t=0}^\infty \gamma^t \mathbb{P}(X_t, X_{t+1}),
which may involve simulating long trajectories when \gamma is close to 1. How is this distribution sampled in practice? Does the sample complexity bound in Theorem 1 account for the cost of generating such samples, particularly in terms of the expected number of steps required per sample?
	3.	Normalization and Adapted Wasserstein:
In Equation (2), the objective \langle \mu, c \rangle lacks an explicit (1 - \gamma) factor. This raises concerns about alignment with the standard definition of the adapted (bicausal) Wasserstein distance, which includes discounting in the cost. Are the authors implicitly including the (1 - \gamma) normalization in the definition of \mu? If so, could this be clarified more explicitly to avoid confusion?
	4.	Connection to Optimal Control and Dynamic Programming:
The adapted Wasserstein distance between Markov chains can be interpreted as the value of an optimal control problem over causal couplings. The variable \lambda_X(y|x) in the paper resembles a policy mapping in this context. Is there a direct connection between the proposed algorithm and dynamic programming or optimal control formulations of adapted OT, such as those explored in Moulos (2021) or O’Connor et al. (2022)? Clarifying this connection could help contextualize the algorithm within the broader literature.

**Ethical Concerns:**

["NO or VERY MINOR ethics concerns only"]

**Final Justification:**

I've read the discussion and responses, I'm satisfied and will keep my positive score.

**Limitations:**

None.

**Paper Formatting Concerns:**

None.

**Quality:**

3

**Strengths And Weaknesses:**

Strengths
	1.	Well-Formulated Estimation Problem:
The paper develops a stochastic algorithm for estimating the adapted (bicausal) Wasserstein distance between two Markov chains from sample streams. The problem is well-posed and connected to classical constructions via occupancy measures and causal couplings.
	2.	Sample-Based Optimization Framework:
By introducing a linear program over normalized occupancy couplings, the authors construct a saddle-point formulation that can be optimized without access to transition kernels. The method—SOMCOT—uses mirror descent and projected gradient ascent, and is supported by finite-time convergence guarantees.
	3.	Sample Complexity Analysis:
The convergence rate in \varepsilon is \widetilde{O}(1/\varepsilon^2), which is consistent with minimax rates for empirical optimal transport. The dependence on 1/(1 - \gamma)^2 is correct under the authors’ normalization, and avoids the more pessimistic 1/(1 - \gamma)^3 rate that arises in standard Monte Carlo estimation of discounted costs.
	4.	Empirical Utility and Side Products:
The dual variables and conditional marginals (e.g., \lambda_X(y|x), \lambda_Y(x|y)) admit interpretations as encoder-decoder maps, and are shown to capture latent structure in block Markov chains. The method also shows promise for model selection and abstraction tasks.

⸻

Weaknesses
	1.	Framing Ambiguity:
Although the paper cites the bicausal OT literature and works with bicausal couplings, it frames the contribution primarily in terms of bisimulation metrics. This obscures the broader significance of the estimator as a sample-based method for adapted Wasserstein distances.
	2.	Per-Iteration Computational Burden:
The per-iteration cost is O(|X|^2 |Y|^2), due to updates over full occupancy tensors. This limits scalability and raises the question of whether dynamic-programming-based or low-rank approximations could yield more efficient alternatives.
	3.	Missing Normalization Clarification:
The paper does not explain that the discounted cost is normalized by (1 - \gamma) in the definition of occupancy measures, and that this normalization is what permits the 1/(1 - \gamma)^2 rate. Without this, readers familiar with standard cost accumulation may incorrectly expect a 1/(1 - \gamma)^3 dependence.
	4.	No Discussion of Known Low-Rank Cases:
In the special case where both chains have rank-1 transition matrices, the problem reduces to static OT between stationary distributions. This is a classical LP (packing type), and the paper does not compare its approach or complexity with recent optimal running times for optimal transport.
	5.	Absence of Lower Bounds:
The paper asserts that its rate is optimal (up to logarithmic factors), but does not prove a matching lower bound or explore dimension-dependent regimes where structural assumptions (e.g., rank, sparsity) could yield improved complexity.

---

> ### Author Rebuttal · Authors · 2025-07-29
>
> Thank you for your careful reading of our work, and a detailed appraisal of the strengths of the contribution! Your comments listed under “weaknesses” are also well-taken; we respond to these below.
>
> 1. **Framing ambiguity:** We opted to mainly frame our contribution in terms of bisimulation metrics as we felt that repeatedly writing out all potential names of the distance (“bicausal OT distance”, “adapted OT distance”, …) would be cumbersome for the reader. We will revise the text to emphasize the connections to (computational) optimal transport more explicitly in the final version.
> 2. **Per-Iteration Computational Burden:** We agree that the quadratic scaling of the runtime with the domain sizes does not look appealing at first sight. That said, we are not aware of any alternatives with better computational complexity, even for the case of known transition kernels. For instance, the Sinkhorn value iteration method of Calo et al. (2024) has the same runtime complexity. We are not aware of any other method with an explicit complexity guarantee. We believe that speedups should be possible under some more concrete assumptions on the kernels (e.g., low-rank), but we did not explore this question in depth so far.
> 3. **Missing Normalization Clarification:** You are absolutely right, our definitions involve a normalization constant of $(1-\gamma)$, and we probably didn’t do a good job at emphasizing the impact of this scaling factor on the rates. We will provide additional discussion in the final version.
> 4. **No Discussion of Known Low-Rank Cases:** We did not provide a detailed review of computational OT to keep the paper more focused (and fit the page limit), but we otherwise agree that a discussion should be added. We will do so for the final version. For now, we just note that one can specialize our method to the “flat” OT problem without Markovian structure, and achieve a rate of order $n^2/\varepsilon^2$ which matches the best rate that we are aware of for Sinkhorn’s method (Dvurechensky et al., 2018) — while requiring only sample access to the distributions! We are not aware of other methods achieving a faster rate. (E.g., naively combining the state of the art method of Blanchet et al. (2018) with a uniform-convergence argument to account for sampling errors would still result in a $n^2/\varepsilon^2$ rate according to a quick back-of-the-envelope calculation.) We felt that this question was a bit niche to explore in depth in the present paper, but we are open to including some additional content in the final version.
> 5. **Absence of Lower Bounds:** We agree that it would be interesting to pin down the exact minimax rates for the unstructured case we consider in this paper, or explore additional assumptions under which faster rates could be possible.
>
> While we agree that all these questions are very interesting (especially the last two), we also believe that our contribution is strong enough on its own right and the paper is quite well-rounded as it is to be worth publishing.
>
> Regarding your additional questions:
> 1. **Comparison with Calo et al. (2024):** Indeed our LP is very similar to the one proposed by Calo et al. (2024), the main difference being that our constraints directly feature the marginal occupancy measures of the two chains. This allows us to develop a primal-dual stochastic optimization method that only requires samples from the occupancies. In contrast, primal-dual methods derived from the previous LP formulation would require sampling from the current estimate $\mu_k$ of the joint occupancy coupling, and querying the transition kernels at these sampled states. This is entirely impractical when having access to trajectory data only. The only possible alternative we can think of would be a simple “plug-in” approach: build an empirical estimate of the two transition kernels based on all observations, and compute the distance between the resulting estimates using a state-of-the-art solver (e.g., Sinkhorn value iteration by Calo et al., 2024). It is not clear at all how the estimation errors would propagate through the analysis; the only analysis we can think of would require the sampling probabilities to be bounded away from zero in every state, and the inverse of the lower bound on the probabilities would appear in the sample complexity bounds. In any case, the resulting method would have the same runtime as our newly proposed algorithm. We are happy to develop this point further in the final version.
> 2. **Sampling Cost under Discounting:** Theorem 1 states the sample complexity in terms of i.i.d. samples taken from the occupancy measures, and does not account for the cost of generating such samples. This is done in Appendix B.4. We believe that it should be possible to use all samples between resets and obtain an improved complexity bound, but we did not work out the details (which might be challenging due to the dependence between the samples).
> 3. **Normalization and Adapted Wasserstein:** Thank you for catching this! We will make sure to include the normalization constants consistently everywhere in the definitions and the derivations (and in particular add the missing normalization constant to the first definition of Eq. 1).
> 4. **Connection to Optimal Control and Dynamic Programming:** Our LP formulation is directly tailored to optimization via stochastic primal-dual methods, which obscures the connection to optimal control a little bit. The LP (and the entire approach) of Calo et al. (2024) makes the connection between the OT problem and the optimal control problem more explicit; we will point this out to the reader in the final version to facilitate navigating the literature.

---

> > ### Comment · Reviewer_52pg · 2025-08-01
> > **Thanks**
> >
> > Thanks for the responses. I maintain my positive view of the paper.

---

> > > ### Author Response · Authors · 2025-08-05
> > >
> > > Thank you for your response & your appreciation of our work!
> > > Best regards,
> > > Authors

---

### Official Review · Reviewer_gruX · 2025-07-14

**Clarity:** 4
**Significance:** 2
**Originality:** 3
**Rating:** 5
**Confidence:** 2

**Summary:**

Calo et al. (2024) showed that bisimulation metrics are a kind of optimal transport (OT) distance.

The present paper extends this by deriving a new stochastic optimization algorithm (SOMCOT) for estimating bisimulation metrics between Markov chains based on sample observations, without knowledge of the transition law. Key to this is proposition 1, which offers a new characterization of occupancy couplings, opening up a novel path to optimization through stochastic primal-dual methods.

Theoretical guarantees are provided and the method is demonstrated on several toy examples

**Questions:**

I would certainly be willing to increase my evaluation if some or all of the following could be addressed. They are all essentially related to practical utility. My feeling is that this work may be useful to the community, and so worth publishing, but that it doesn't make this case clearly enough at the moment.

1. Better comparison with existing methods

I wouldn't mind if the example is very small, but I do think there should be at least some way of comparing the current approach to previous methods, even just to see what is lost in terms of accuracy moving to a sample-based approach from explicit transition kernel knowledge. Would it be possible to use the approach of Calo et al. (2024) or maybe Chen et al. (2012)? I'm not particularly familiar with those works, but there must be some other method that could be compared to. I'd be very happy to see competitive accuracy from SOMCOT and perhaps demonstrating a regime where SOMCOT is superior.

2. Empirical scalability

How does performance degrade on practical problems when we are limited in computation?

3. Could the authors elaborate on a plan towards extending the approach to infinite state spaces using function approximation etc.? This isn't vital, but if they can show anything relating to it then that does strengthen the claim that the paper is a foundational step into a promising direction, and not a dead end.

**Ethical Concerns:**

["NO or VERY MINOR ethics concerns only"]

**Final Justification:**

On reflection, I take the authors' point that the work is quite original and I was unfair in my original assessment that they ought to compare it more thoroughly to other approaches. I would appreciate if they could spare a few more words to this effect in the final paper, as it was unclear to me. The work is overall very strong and I have a more positive outlook on its merits having read their response. Accordingly, I've upgraded my score.

**Limitations:**

yes

**Quality:**

4

**Strengths And Weaknesses:**

Strengths

- This is a sophisticated contribution with extensive theoretical guarantees. It seems to me a rigorous and interesting piece of work. It's also well written.
- The approach relies only on sampling and not on knowledge of the transition law. This relaxation is a meaningful step forward and non-trivial.
- A key contribution is the new LP formulation in proposition 1 which, if I’ve understood it correctly, I find to be a clever result allowing us to describe occupancy couplings without using the transition kernel explicitly. It does so by introducing auxiliary variables representing conditional distributions under the optimal coupling.
- The paper offers an analysis of sample complexity with explicit constants.

Weaknesses

- There are at least two clear limitations to the work: it relies on finite state spaces and is extremely computationally intensive.
- The finite state space assumption. While the authors acknowledge this limitation, it really does limit the applicability of the methods and there is little in the way of evidence to suggest an extension is possible.
- The computational complexity. The sample complexity [O(|X||Y|(|X| + |Y|)/eps^2)] is clearly highly undesirable, as is the cost per iteration [Theta(|X|^2 |Y|^2)]. This implies for example that in a fairly modest system of 100 states, you would need ~2M/eps^2 samples, which for eps=0.1 (a low bar in some problems) implies 200M samples. I would assume, though I’m no expert, in realistic reinforcement learning scenarios, this will balloon to trillions of samples.
- As far as I can see, the paper does not compare the approach empirically to any other existing method. I take the authors’ point that this method is the first of its kind (p.2) in some respects, but without any kind of comparison it’s hard to say whether the algorithm offers any advantage over simpler baselines, or how its complexity compares in practice to alternatives. They could have restricted their attention, at least in one example, to a case where this comparison was possible, e.g. some tractable small example where many methods could be used. The lack of empirical comparison is an issue: (i) exact methods exist in some small state space examples, (ii) I cannot tell from this work when to prefer this method over alternatives that may be available.
- Nit: is it standard in this field to use x bar to mean an infinite-length vector x? In statistics this commonly refers to a sample mean, so this seems a little confusing.

---

> ### Author Rebuttal · Authors · 2025-07-29
>
> Thank you for your careful reading and thoughtful comments on our work! We are glad to see that you have appreciated the originality of our contribution and the rigor of our theoretical analysis. We also acknowledge the criticism that our approach is currently limited to finite state spaces, and that it is not entirely obvious at this time how to scale the approach to problems with high-dimensional state spaces. We would nevertheless like to push back on some of the concrete points made in your review, hoping that we can reach agreement about the value and potential of our work.
>
> First of all, we would like to double down on our point that our method is “the first of its kind”: we truly are the first to rigorously consider the problem of estimating bisimulation metrics / OT distances between Markov chains, and therefore there are literally no other known methods that aim to solve the same problem. Even for small problems, there are no exact methods or known alternatives. The only possible alternative we can think of would be a simple “plug-in” approach: build an empirical estimate of the two transition kernels based on all observations, and compute the distance between the resulting estimates using a state-of-the-art solver (e.g., Sinkhorn value iteration by Calo et al., 2024). This approach has no guarantees on the estimation error, and has the same runtime as our newly proposed algorithm. We are happy to make this point more explicit in the final version.
>
> Second, regarding the scaling with the sizes of the state spaces: we of course agree that a quadratic scaling does not look appealing at first sight. Nevertheless, occupancy couplings (the key component in the *definition* of our distance metric) are of size $|\mathcal{X}|^2|\mathcal{Y}|^2$, and thus it is clearly unrealistic to expect that such an object can be computed in sub-quadratic time. Just constructing the required data structure costs $|\mathcal{X}|^2|\mathcal{Y}|^2$ time and space. In light of this, we actually found it quite surprising that we could achieve a sample complexity guarantee of the improved order $|\mathcal{X}||\mathcal{Y}| (|\mathcal{X}|+|\mathcal{Y}|)$, and we believe that this guarantee is not improvable in general. Of course, improvements might be possible under further structural assumptions (e.g., if the transition kernels are low rank as another reviewer has suggested), but we feel that such extensions are well beyond the scope of this paper.
>
> Finally, about the assumptions about the finiteness of the state spaces and the potential to scale up our algorithmic ideas: these are indeed the questions we are planning to address next (as mentioned in Section 6). While you are correct to point out that the method is unsuitable for large state spaces in its current form, we have serious reasons to believe that it can serve as an important foundation for a more practical method that can deal with larger problems. In particular, since the algorithm is based on gradient descent (and not dynamic programming like previous methods for computing OT distances), there is a natural way to combine it with “function approximation”:  just parametrize the primal and dual variables and perform stochastic gradient descent/ascent on the resulting parametric approximation of the Lagrangian. Doing this extension well enough will require some further careful work that is beyond the scope of the present work.
>
> Once again, we would like to recall that our work is the very first to provide theoretical guarantees for this problem setting. Our objective was to develop a robust set of foundational ideas and to perform a careful and complete mathematical analysis in the setting we have considered. We believe that these foundations will be very useful for further developments, and that many colleagues in the NeurIPS community will find them inspiring.

---

### Decision · Program_Chairs · 2025-09-17

**Decision:**

Accept (poster)

**Comment:**

This paper discusses the problem of estimating bisimulation metrics (equivalently, adapted/bicausal Wasserstein distances) between Markov chains using only sample trajectories rather than full transition dynamics. The authors develop a stochastic primal-dual algorithm that they term SOMCOT, a linear programming (LP) reformulation that sidesteps the need for explicit transition kernels. The work is technically strong with finite-sample guarantees, though currently restricted to finite state spaces with quadratic computational complexity.

__Strengths__:
- First method to estimate these distances from samples alone - previous work required full transition dynamics
- Novel LP formulation over occupancy couplings enables stochastic optimization
- Tight sample complexity bounds with rigorous theoretical analysis
- Clear writing and mathematical presentation
- Connects optimal transport with RL bisimulation metrics

__Weaknesses__:
- $O(|X|²|Y|²)$ per-iteration cost limits scalability
- Restricted to finite state spaces with no clear extension path
- All experiments synthetic, no real-world demonstrations
- Millions of samples needed even for modest-sized problems

The consensus leans toward acceptance as a solid theoretical contribution that opens a new research direction, despite practical limitations that future work must address.